computational biology, ecology, health and disease and epidemiology

measles, spatial epidemics, infectious disease dynamics, population dynamics, cross-scale dynamics

**Author for correspondence:**
Alexander D. Becker
e-mail: adbecker@princeton.edu

# Coexisting attractors in the context of cross-scale population dynamics: measles in London as a case study

Alexander D. Becker[1], Susan H. Zhou[1], Amy Wesolowski[2] and Bryan T. Grenfell[1,3,4]

[1]Department of Ecology and Evolutionary Biology, Princeton University, Princeton, NJ 08544, USA
[2]Department of Epidemiology, Johns Hopkins Bloomberg School of Public Health, Baltimore, MD, USA
[3]Fogarty International Center, National Institutes of Health, Bethesda, MD, USA
[4]Woodrow Wilson School of Public and International Affairs, Princeton University, Princeton, NJ, USA

ADB, 0000-0001-6524-0513; AW, 0000-0001-6320-3575

Patterns of measles infection in large urban populations have long been considered the paradigm of synchronized nonlinear dynamics. Indeed, recurrent epidemics appear approximately mass-action despite underlying heterogeneity. However, using a subset of rich, newly digitized mortality data (1897–1906), we challenge that proposition. We find that sub-regions of London exhibited a mixture of simultaneous annual and biennial dynamics, while the aggregate city-level dynamics appears firmly annual. Using a simple stochastic epidemic model and maximum-likelihood inference methods, we show that we can capture this observed variation in periodicity. We identify agreement between theory and data, indicating that both changes in periodicity and epidemic coupling between regions can follow relatively simple rules; in particular we find local variation in seasonality drives periodicity. Our analysis underlines that multiple attractors can coexist in a strongly mixed population and follow theoretical predictions.

## 1. Introduction

A central question in ecology is how simple, often periodic, effectively mass-action dynamics can emerge at the aggregate population level despite potentially complicated mechanisms, heterogeneities and interactions at finer spatial scales [1–3]. Dynamics of oscillatory host–natural enemy interactions are one such canonical example. Childhood infectious diseases have proven to be an excellent testbed to examine how finer-scale processes relate to, and inform, macro-scale dynamics [4–6]. In particular, measles, due to its simple natural history and long-time series of data of the pre-vaccination incidence from the US and UK, has provided a rich source to understand how dynamical patterns aggregate at various spatial scales [7–9].

An array of dynamics ranging from regular multiannual infection patterns to deterministic chaos have been disentangled using well-mixed epidemic models [8–11]. Birth rate and seasonality in transmission in particular have been shown to be major drivers of periodicity at the large city level of epidemics [7,9,11–15]. Seasonality in measles appears to arise from the aggregation of human populations. In the US and UK, seasonal transmission patterns have typically mirrored school calendars, with increases in transmission at the start of school terms [9,10,16,17]; other movement-related drivers have been implicated in sub-Saharan Africa [11]. In a US city-level analysis, [9] subtle changes in contact structure led to drastic differences in periodicity, suggesting that regional variation may be important. Despite this rich heterogeneity in observed dynamics, analyses have often focused on urban populations in relative isolation at, or above, the critical community size (CCS; the population size required to maintain endemic

transmission) of approximately 200 000 due to the measles patterns observed and their associated predictability [10,18,19]. These analyses typically assume that epidemic differences may be insignificant between sub-populations, instead supposing that multiple interactions at finer spatial scales can be well approximated by the dominant periodicity of the overall population. Epidemic coupling *between* urban populations has been studied both theoretically and via rich metapopulation datasets, captured by gravity models [4]. These studies emphasize how changes in coupling rates can produce an array of in- and out-of-phase dynamics, and seasonal forcing can help maintain (a)synchrony [20,21]. However, few analyses have examined questions of synchrony and the presence of coexisting attractors at the within city scale; this gap arises largely from a lack of data. In this paper, we use a set of recently digitized data from London to identify evidence of dynamic heterogeneity across multiple spatial and population scales.

Analysing borough-specific measles mortality records from pre-vaccination London (1897–1906), we find that, although homogeneous mixing produces dynamics consistent with the annual patterns seen at the aggregate level, clear annual and biennial cycles can be observed when examining the finer-scale data. Epidemiologist John Brownlee first noticed this pattern in 1917, postulating that 'varying local conditions in different parts of the city may be sufficient' to produce the array of periodicities found in the London regions [22]. Although Dr Brownlee analysed the cross-scale London data using periodograms (a method still crucial in the field [23]), his approach, based on a pathogen's time-varying infectivity, differed from the mechanistic framework now commonly used in the modern era [24,25] and necessary to estimate seasonal transmission parameters and test the role of demography on epidemic dynamic.

We use recently developed statistical methods combined with a stochastic epidemic model to determine both the drivers of varying periodicity and the coexistence of multiple attractors in the same system. We uncover key differences in regional seasonality correlating with underlying periodic dynamic theory as well as pairwise coupling, allowing for accurate predictions of epidemic timing and periodicity. These results provide new evidence for how relatively simple patterns can emerge, and coexist, in nonlinear ecological dynamics.

## 2. Methods: data, mechanistic model, inference framework and underlying theory

We analysed measles mortality records across the five inner and an approximate four outer regions (e.g. the Outer Ring) of London from 1897 to 1906 (figure 1). Although the Outer Ring of London was not formally divided into regions, we grouped them as outer East, West, North and South, based on their present-day geographic coordinates and nearest inner region. Weekly mortality records in which the cause of death is listed as measles, annual population counts and yearly birth estimates per region were digitized from the Registrar General's reports [26]. The mortality data were aggregated to an approximately monthly (i.e. four weeks) scale in order to amplify the periodic signal. Yearly birth and population data were interpolated to the monthly scale to smooth parameter estimates. Measles periodicity was quantified by taking the nearest-integer period corresponding to the maximum peak in the spectral density [23]. The time series data for each location are included in the electronic supplementary material.

We used a stochastic susceptible–exposed–infected–recovered (SEIR) model with a sinusoidal seasonal transmission rate, $\beta(t) = \overline{\beta}(1 + \alpha \sin(2\pi t + \phi))$, where $\overline{\beta}$ is the average transmission rate, $\alpha$ and $\phi$ the seasonal amplitude and phase, to analyse deviations in periodicity between regions. To estimate the time-invariant case fatality rate (CFR) [27], we assumed perfect reporting of measles mortality, hence this is somewhat of a relative metric in the case of imperfect measurement. Additionally, we estimated a mean importation rate, $\iota$, for each region to prevent stochastic extinction in the model. To avoid overfitting to the monthly data (approx. 125 data points), we fixed the infectious and latent periods at 5 and 8 days, respectively [28]. Using the iterated filtering algorithm, we fit the stochastic SEIR model to each region using an observation process that allows for measurement error per [13]. Briefly, the iterated filtering method aims to maximize likelihood by allowing all parameters of interest to take random walks in tandem. The variance of these walks is slowly reduced, enabling the algorithm to both overcome any local valleys in the likelihood surface, and converge to the optimal value. This process is repeated many times (here, we used 400 unique parameter sets) to acquire estimates of the likelihood for each parameter set, as well as fully explore the multi-dimensional parameter space for 60 iterations for each parameter set. A full discussion of the algorithm, and numerical implementation, can be found in [29,30] with more details in the electronic supplementary material. Once parameters are obtained, we forward simulated our fitted model to compare against the data both visually and in terms of power spectra.

To provide a theoretical basis for how variation in seasonality affects periodicity, we performed a simulation study. Using the same seasonal transmission function, $\beta(t) = \overline{\beta}(1 + \alpha \sin(2\pi t + \phi))$, we fixed $\overline{\beta}$ equal to the mean inferred transmission rate for the nine regions. Using a range of amplitude (range: 0.03–0.35) and phase values (range: 0–2, corresponding to a maximum lag of approximately four months), we numerically integrated a deterministic SEIR model (equations shown in the electronic supplementary material) for each set of parameter values to produce a set of theoretical predictions (figure 3b). For each simulation, we quantified the periodicity by taking the nearest-integer period corresponding to the maximum power via spectra analysis.

To investigate how variation in regional periodicity impacts the spatial spread of infection within London, and the role of any potential regional coupling on periodicity, we constructed a stochastic two patch SEIR metapopulation model. While we analysed each region in isolation above, we relaxed this assumption and allowed an interaction between two locations. For each location, we modulated the force of infection to include a coupling parameter related to the number of infected in the other location ($\beta(I + \iota)/N$ to $\beta((I_X + \iota_x)(1 - \sigma) + I_Y\sigma)/N_x$ for population X coupled with population Y, where $\iota_x$ refers to the importation rate into population X). This coupling parameter, $\sigma$, represents the strength of coupling or the relative contribution between the two populations [21]. For example, a coupling rate of 0.01 between locations $X$ and $Y$ is equivalent to assuming that the infected population in $Y$ contributes 1% of the new infections, albeit scaled by $\beta/N$, in population $X$. To parameterize the two-region model, we used the parameters inferred from the region-specific monthly data (e.g. seasonal transmission values) with the goal of estimating pairwise regional coupling. In order to simplify the analysis, we used the West region as a

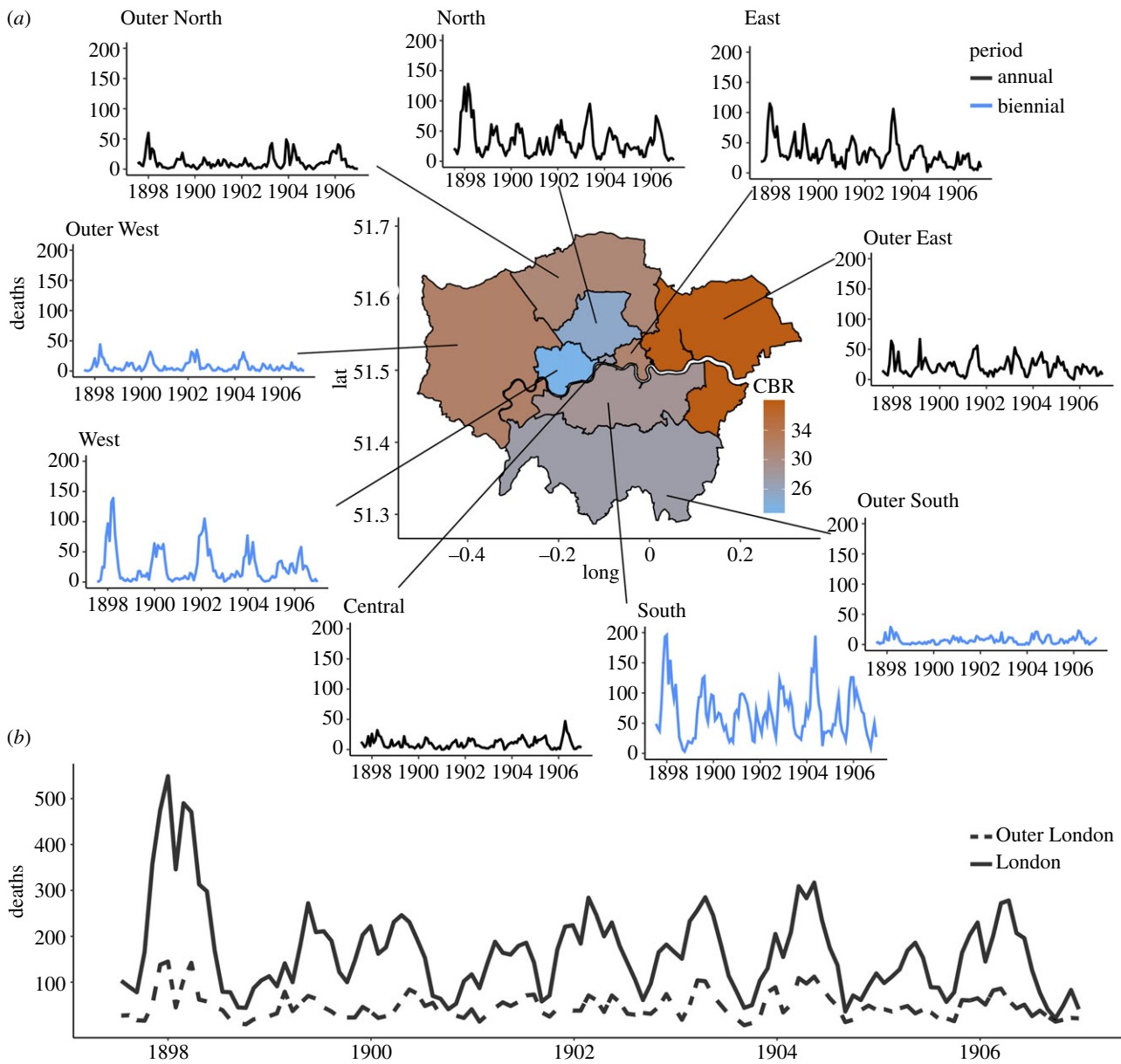

**Figure 1.** Time series of measles dynamics in London. (*a*) Monthly measles mortality reports from the five inner and four outer regions in London, 1897–1906. Each region is coloured by its crude birth rate (CBR) ranging from 23 (in the West) to 38 (in the outer East). (*b*) Monthly measles mortality reports for the aggregate city of London (solid line) and the Outer Ring (dashed line) for the same time period. Each time series is colour coded by the region's periodicity. (Online version in colour.)

case study due to the region's firmly biennial pattern (the same analysis for the annual East region is shown in the electronic supplementary material). Assuming local parameters such as seasonality, importation rate and initial conditions can be inherited from the regional analysis above, we then estimated a coupling value by profiling over the range of plausible values (zero, indicating completely separate populations to 0.5, indicating equal mixing) and selecting the maximum-likelihood value, again using the iterated filtering algorithm, for the variable periodicity North–West pair and the both biennial West–Outer West pair. We compared the inferred coupling rates with the observed dynamics in those regions by simulating the metapopulation model across the range of coupling to construct probability distributions.

## 3. Results

From 1897 to 1906, London's population was 4.5 million with an average crude birth rate (CBR = number of births in a year ÷ the

mid-year population × 1000) of 29 (range: 28 to 30); divided into five regions (figure 1). Outside London (i.e. the Outer Ring), consisted of an additional 2 million individuals across an approximate four outer regions. The most populous region (South) had a population of nearly 1.7 million whereas the smallest (Central) only had a population of 270 000. Importantly, all regions were above or near the CCS for measles (estimated at 200 000), suggesting that it was possible for measles to endemically circulate. This was supported in the data in that each region contained few, or zero, months of no measles mortality. Crude birth rates were also highly variable across London and the Outer Ring, ranging from 22 to 52 across the 10 years (average CBR for each region shown in figure 1).

Using a power spectral analysis, we found that although the aggregate city-level mortality data was firmly annual, there was region-level heterogeneity in periodicity. The North, East and Central regions were also annual; however, the West and South were biennial (data shown in figure 1; power spectrum shown in the electronic supplementary material). The Outer Ring mimicked this pattern with outer North and East being

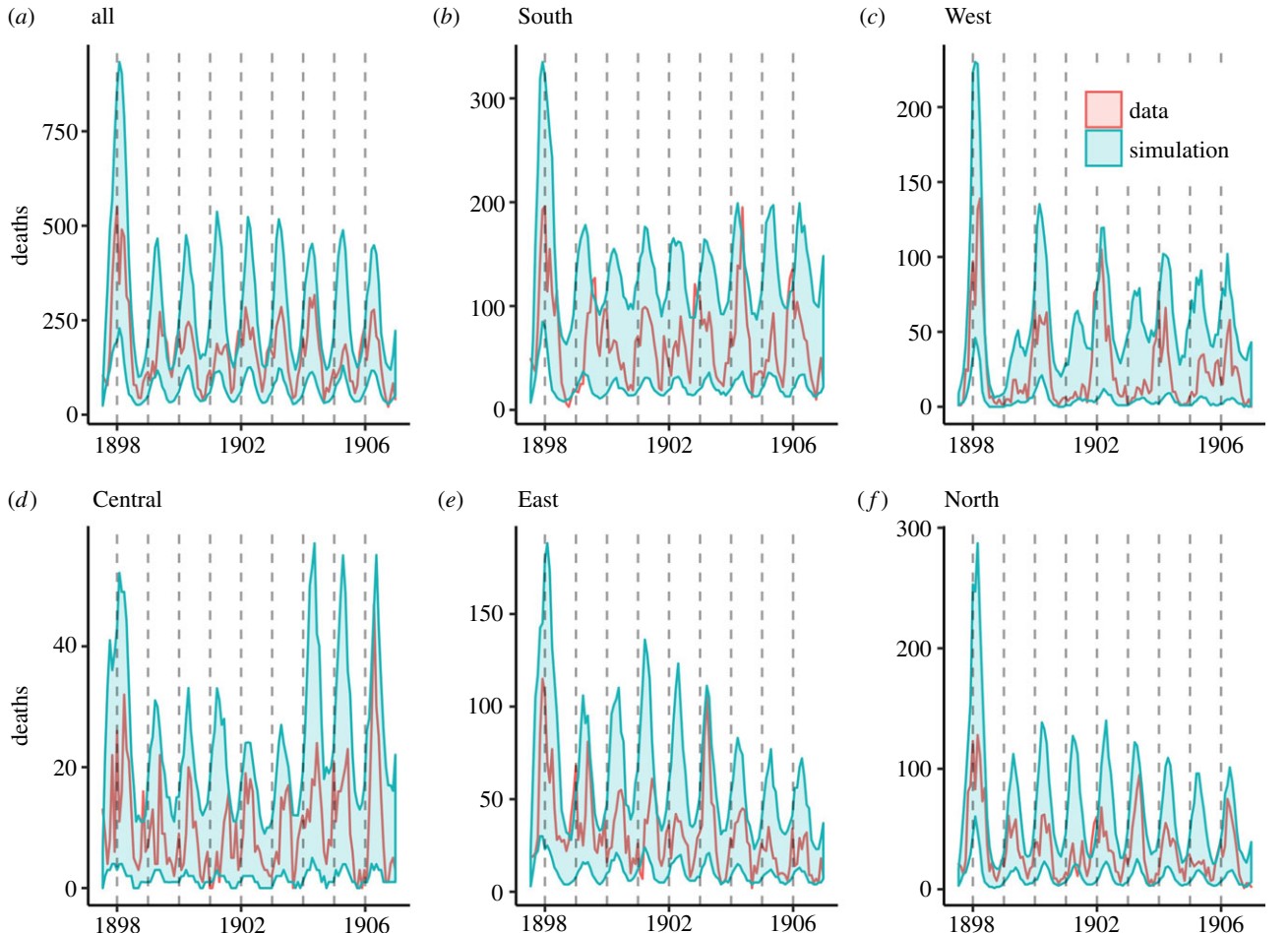

**Figure 2.** Observed and predicted patterns of measles in inner London. Simulating the fitted SEIR model yields generally close fits to the aggregate and region-level data. The simulations, shown in light blue bands of 95% quantiles, generally visually match the observed data, shown in red, in terms of periodicity, outbreak size (particularly the 1898 outbreak) and general shape. The outer region fits are shown in the electronic supplementary material. (Online version in colour.)

annual, while outer South and West were again biennial. However, some regions, in particular Outer South and Outer West, were noisy, possibly due to overall lower mortality counts and therefore potentially limiting our ability to infer estimates of periodicity. Surprisingly, the large variation in CBR did not correlate with the observed periodic signal.

The simple seasonally forced stochastic SEIR model yielded a generally well-matched visual fit when calibrated against the data to these heterogeneities across individual locations, shown in figure 2 for the inner regions (fits for the Outer Ring are shown in the electronic supplementary material). In particular, we note the fitted model was able to visually capture the periodicity, in particular the West region, and the general shape and size of epidemics across locations, in particular the larger 1898 outbreak. We further compared the predicted power spectra against the data (shown in the electronic supplementary material). While the majority of regions agreed well with the observed spectra, some such as the South and outer West departed from the observations, indicating that in those regions not all stochastic realizations of the fitted model were biennial. In addition to visual fit, the model captured key biological parameters. For example, our estimated CFR (ranging from 0.9% to 1.7%) compared well with a meta-analysis by [27], in which the observed nationally representative CFRs fell between 0.23 and 1.51%. Early 1900s estimates of CFR for London are challenging to assess as measles cases were not yet notifiable and therefore the incidence reporting rate is unknown for this era; however, the reported CFR for Paddington from

1904 to 1906 was 2.7%–4.3% [31]. Crudely applying the incidence reporting rate of approximately 50% in the later era [13] yields an approximate CFR of 1.35%–2.15%, in line with our estimates here. However, we assumed complete reporting accuracy in both our model and historical rate correction. In the case of imperfect reporting, the estimated CFR would probably increase. All parameter estimates, as well as population sizes, are shown in the electronic supplementary material.

As hypothesized a century ago [22], local variation in seasonal transmission patterns appears to have been the main driver of measles periodicity within London; both seasonal amplitude, $\alpha$, and phase, $\phi$, are well captured by the models. In particular, the biennial regions (e.g. West and South) exhibited lower amplitudes, while also showing greater phases, than their annual counterparts (North, East, Central and London aggregate), as shown in figure 3$a$, colour coded and grouped by periodicity, with the seasonality of each sub-region shown as a point, and the group mean (+/− standard error) shown as a line. The aggregate seasonality patterns are shown as squares in figure 3$a$ and $b$ and not included in the mean seasonality curves. Analysing this pattern further, we constructed a dynamic diagram based on these two-dimensional transmission parameters while using the mean inferred transmission value across the nine sub-regions as a baseline (mean $R_0 = 22$, range 19–25). Numerically integrating the deterministic SEIR model over a range of amplitudes and phases as part of a qualitative analysis, we found basins of both annual and biennial dynamics (figure 3$b$). In the low-amplitude regime,

Proc. R. Soc. B 287: 20191510

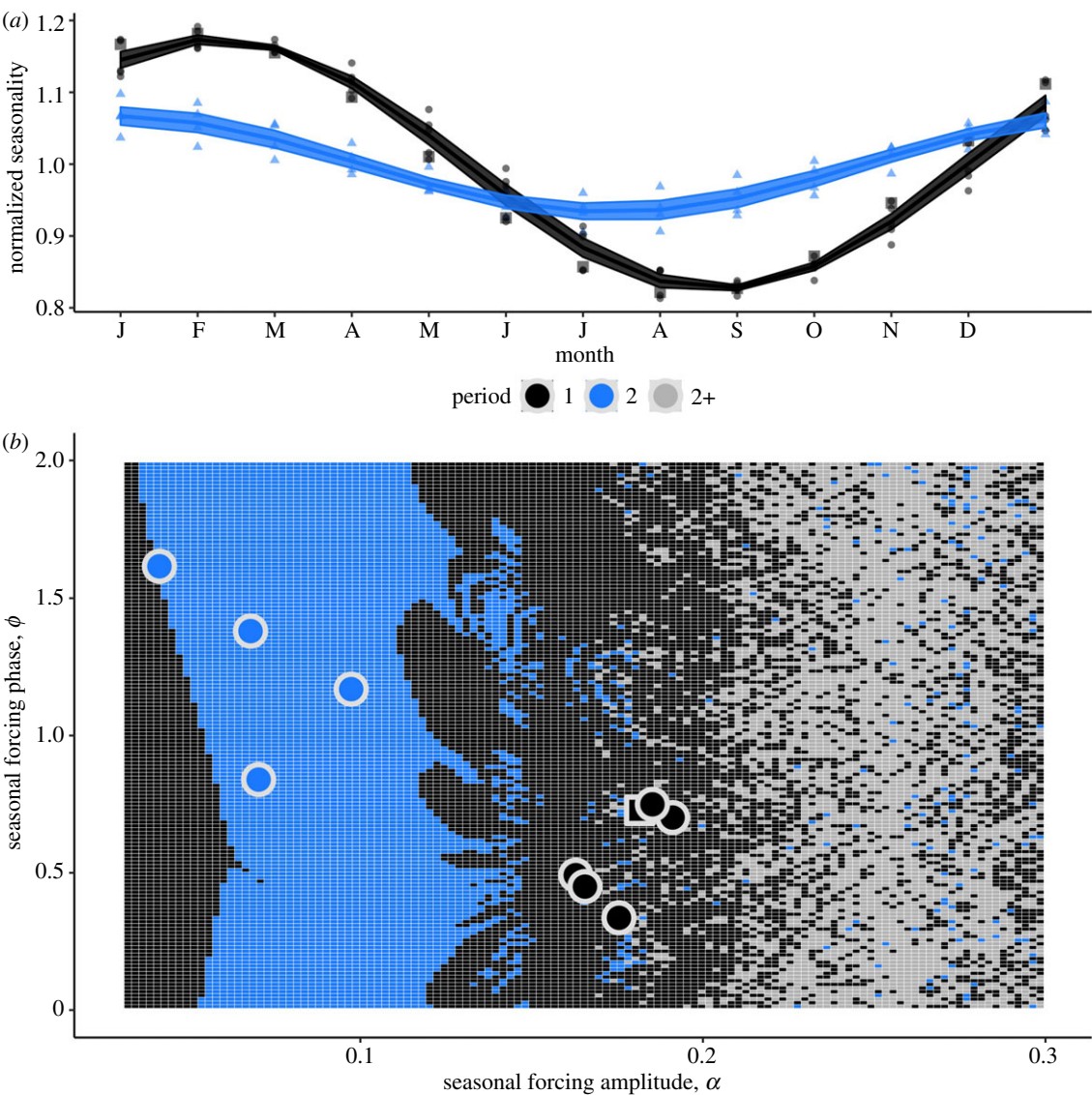

**Figure 3.** Dynamic consequences of changing seasonality patterns on measles periodicity in historical London. (*a*) Normalized, inferred seasonal transmission for each region (points) grouped and coloured by their periodicity (curve refers to group mean +/− standard error) in London. The inferred aggregate seasonality pattern is shown as square points. (*b*) Dynamic diagram for a deterministic, seasonally forced SEIR model. Seasonality is modelled by the same sine function as in the regional inference: $\beta(t) = \beta\left(1 + \alpha \sin(2\pi t + \phi)\right)$. The *x*-axis, the forcing amplitude, $\alpha$, and the *y*-axis, the forcing phase, $\phi$, dictate the periodicity, with 1-, 2- and >2-year cycles shown in blue, black and dark grey, respectively. The circles are the inferred seasonal values for each London region, colour coded by the observed periodicity in the data. The phase and amplitude of the aggregate location are shown as a square point. (Online version in colour.)

we observed biennial patterns. Moving into a moderately forced regime (0.1–0.2), annual dynamics began to dominate. When forcing becomes greater than 0.2, we observe an array of dynamics, where 1-, 2- and >2-year cycles are possible. Overlaying the inferred amplitude and phase seasonality patterns (values shown as points in figure 3*b*), we found a qualitative agreement between the data and theoretical forecasts. Generally, the inner and outer regions fall into their predicted attractor basins, as do the aggregate city data (shown as a square point), yielding a plausible explanation for how seasonality patterns drive periodicity even among spatial heterogeneity. Indeed, although birth rate is not unimportant, we found seasonal amplitude and phase to both better explain the observed variation in periodicity (principal components analysis shown in the electronic supplementary material).

Thus far we've analysed local dynamics to gain insight into the drivers of region-specific measles periodicity; we now turn to the potential role of population coupling on producing

metapopulation dynamics of the same periodicity between two regions. At first sight, the relatively strong coupling in cities would promote each region producing the same periodic signal; however, a body of theoretical work has emphasized that seasonal forcing can maintain variable periodic signals and asynchronized dynamics at even high coupling rates [20,21]. Here, we explore this tension between coupling and local heterogeneity through a novel data-driven approach. We hypothesize that the observed regional differences in seasonality may allow attractors to remain in differing periodicities, possibly due to modulating the effective $R_0$ at the start of each year. To test this hypothesis, we examined a two-patch model using West London as a case study. West London (biennial dynamics) was chosen because of its close geographic proximity to the North region (annual) and Outer West (biennial), two neighbouring regions with different dynamics. For each pair, we simulated the two-patch epidemic model over the full range of potential coupling values (ranging from zero,

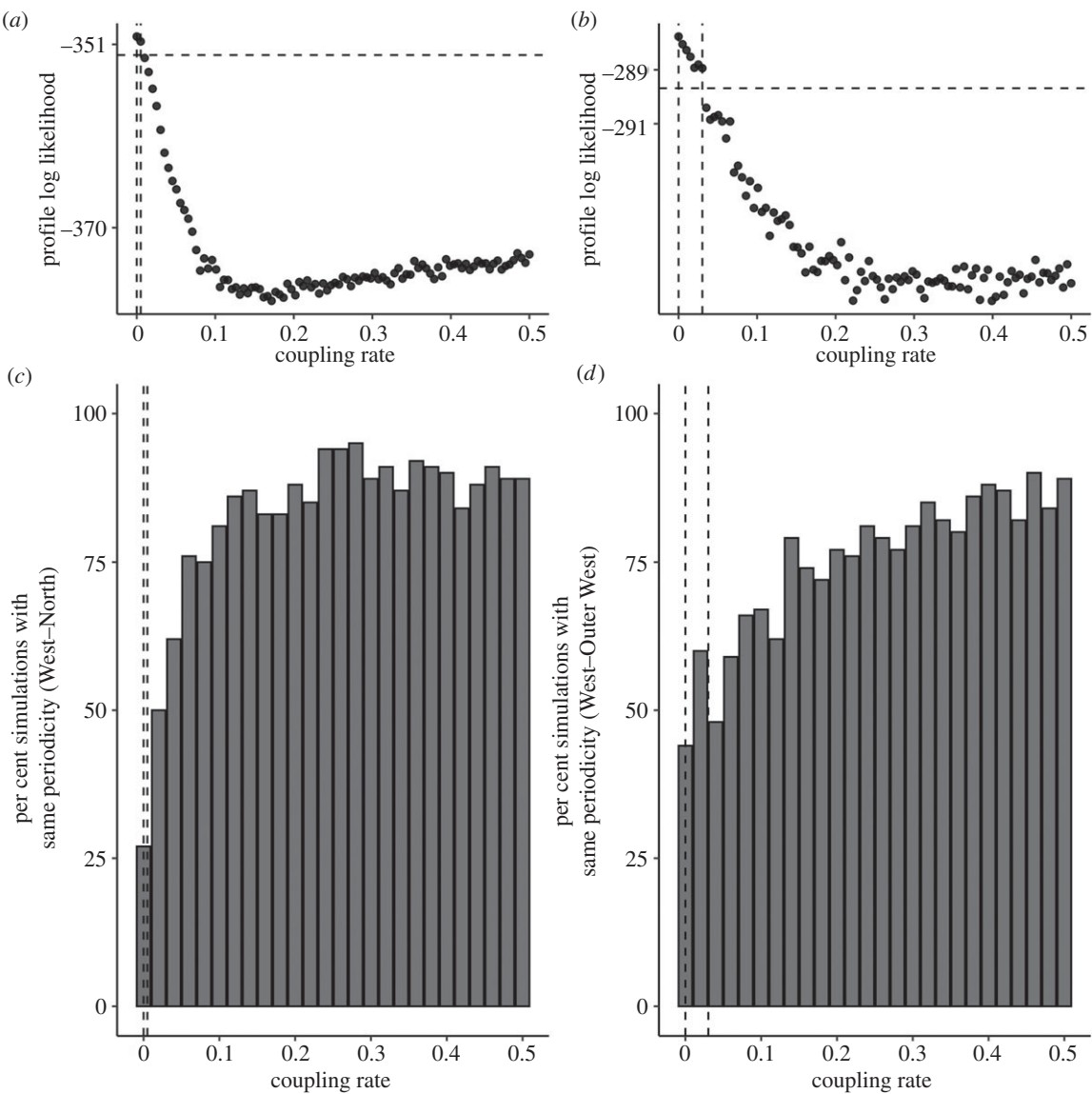

**Figure 4.** The impact of spatial coupling on outbreak dynamics using West London as a case study. (*a*) The estimated coupling rate between the annual North and biennial West regions, where dashed lines indicate the 95% confidence intervals. (*c*) The per cent of simulations which resulted in outbreaks with the same periodicity for each coupling rate between North and West London. The two dashed vertical lines refer to the estimated coupling rate inferred from the pairwise model. (*b,d*) The same analysis, but for the biennial West and biennial Outer West regions. (Online version in colour.)

corresponding to complete isolation up to 0.5, corresponding to a well-mixed population). For each simulation, we quantified the periodicity in each *in silico* region. For the annual North and biennial West pair, we found low regional coupling explained the observed difference in periodicity (figure 4*c*). By contrast, most values of coupling gave a greater than 50% chance of the West–Outer West pair producing dynamics of the same periodicity (figure 4*d*), indicating similar dynamics between regions across any range of population coupling.

To confirm these theoretical results, we then estimated the coupling rate using our epidemic model. For the annual North–biennial West pair, the estimated coupling rate (95% CI: 0–0.01) corresponds to approximately a 25% chance of dynamics of the same periodicity, matching well with the observed data (figure 4*a*). Indeed, this parameter is well identified both in terms of fitted parameters and resulting epidemic periodicity as outside of the estimated confidence interval, the chance of equal periodicity dynamics rises quickly above 50%. For the biennial West–biennial Outer West pair, the estimated coupling rate (95% CI: 0–0.03) denotes an approximately 50%

chance of equal periodicity dynamics. The same set of analysis for the East region is shown in the electronic supplementary material. In general, coupling rates across the 14 pairwise regions were low, with the highest rates generally occurring between an inner and outer region (electronic supplementary material). An additional theoretical, deterministic analysis of the role of epidemic coupling on periodic synchrony is shown in the electronic supplementary material. These results add data-driven support to previous theoretical results, and provide evidence for predictability even among rich heterogeneity at spatial subscales.

## 4. Discussion

A prominent question in ecology and epidemiology is how relatively simple dynamics emerge at the aggregate in spite of potentially unsynchronized or variable periodicity dynamics at subscales. However, analyses of such meso-scale events are rare due to the demanding temporal

and spatial grain needed in the data. With their rich datasets and relatively simple clockwork, the dynamics of childhood epidemics provide a powerful testbed for these ideas. Previous work has generally analysed dynamics at the aggregate city-level primarily due to data availability. Indeed, endemic pre-vaccination data at the regional level will likely only be found in a handful of cities. In this light, the 1897–1906 London regional data are unique in both their existence and observed heterogeneity.

The close visual fits to the data shown in figure 2 inspire confidence in the inferred model's ability to accurately describe the underlying dynamics, thus allowing us to perform detailed comparison with theory. Similar to [9], we find that subtle differences in seasonal transmission structure can lead to different periodicities. Using a dynamic grid based on this variation in seasonal amplitude and phase, we provide a plausible explanation for the array of observed data based on measurable differences in seasonality. While long-term deterministic dynamics become invariant to seasonal phase, shorter-term, non-equilibrium dynamics display rich heterogeneity as exhibited in figure 3b. As seasonal phase was identified as a key parameter in a PCA analysis, we chose to model this likely transient-driven dynamic. Future work aims to quantify and better understand these ecological transients sensu [32]. What is most surprising, however, is that these periodicities can coexist and remain constant in such close proximity. This was explored using a pairwise coupling analysis, finding agreement, in particular in the North–West pair, between observed dynamics and estimated coefficients. Although the pairwise analysis provides a plausible and computationally tractable explanation for the observed heterogeneity, it may not discount more complex, higher-order interactions. Additionally, although we assumed a constant importation rate in our local analysis, including a fraction of infected individuals in other populations (i.e. a full metapopulation model) could provide an alternative approach that would allow for the disentanglement of internal coupling and external importations. These processes, and their associated spatial models, are an area for future research.

Our results prompt the general question: what socio-demographic forces drive heterogeneities in epidemiologically relevant mixing? Although the shape of the estimated seasonal transmission explains variation in periodicity between regions, we were unable to find historical documentation, such as regional school calendars, indicating explanatory differences. However, since each inferred seasonality approximately aligns with the school calendar (e.g. peaking in February–March and September–October, with the lowest values in the summer months), we can speculate that regional variation in school may be the underlying mechanism. An obvious driver would be population density [28]; however, we did not find any simple scaling. We did, however, find records documenting more challenging living conditions in the West [33], perhaps indicating that fewer children attended school in this region, therefore impacting the estimated seasonality. However, the similar inferred CFR between regions suggests the model may be unable to distinguish these socio-economic differences. Additionally, the inferred CFRs may be an underestimation given the immunosuppressive nature of measles infection [34]. Likewise, differences in socio-economic conditions may have manifested themselves in the substantial range of crude birth rates observed across the nine regions spanning 10 years (22 to 52); however, this variation, while still important to the overall dynamics, was not identified as a primary driver of periodicity. Further, although the monthly region-level incidence data analysed in this manuscript generally exhibit clear periodicities, albeit some exceptions in the outer regions, mortality data may lag slightly behind the true incidence counts [35], leading to potential biases in estimates of seasonal phase. However, these potential biases are likely to be consistent across locations. Additionally, by assuming a constant CFR we are effectively treating the mortality data as scaled incidence data, although lower count regions, such as the Outer Ring, may present challenges in inferring periodicity from the fitted model (see electronic supplementary material). Combining and analysing tandem case and death data is a future area of research likely to yield novel insights into time-varying case fatality rates as well as potential statistical biases in methodology. Additionally, although the dynamic diagram produced using the mean approximation of transmission yielded a plausible, qualitative explanation of the data, such averages may impact periodicities in subtle ways as well, and more systematic, and higher-order, dynamics should be explored in the future. Finally, it is also worth noting that the much-analysed London dynamics following World War II are strongly synchronized with the same periodicity throughout the city [36]; this may imply an increase in population coupling between regions.

The accelerating interest in digitizing historical medical records has reenergized the use of childhood diseases as a testbed for both theoretical and data-driven analysis [16,37,38]. In addition, recent statistical advances now allow for confronting complex models with data. Future work will aim to examine spatial coupling for polymicrobial infections. Generally, our results indicate that while aggregate-level patterns of transmission may be readily analysed by assuming homogeneity, extensive heterogeneity may exist at subscales. By disentangling cross-scale spatial epidemic data with mathematical models, we have shown that the diversity of observed patterns can be explained using simple underlying theory, furthering our ability to understand drivers of periodicity in ecological metapopulations. Finally, as global and regional populations become even more connected in the modern era [39], our study lends itself to the importance of understanding the role of spatial coupling in epidemic dynamics.

Data accessibility. Data are provided at https://github.com/adbecker/LondonRegionalAnalysis/.

Authors' contributions. A.D.B. and S.H.Z. collected the data. A.D.B., S.H.Z. and A.W. performed the analysis. All authors wrote and edited the paper.

Competing interests. We declare we have no competing interests.

Funding. A.D.B. was supported by a National Science Foundation Graduate Research Fellowship and the Center for Health and Well-being (CHW) at Princeton University. A.W. was supported by a career award at the Scientific Interface from the Burroughs Wellcome Fund and a National Institutes of Health Director's New Innovator Award. S.H.Z. was supported by the department of Ecology and Evolutionary Biology at Princeton and the CHW. B.T.G. was supported by the Research and Policy for Infectious Disease Dynamics program of the Science and Technology Directorate, Department of Homeland Security, the Fogarty International Center, National Institutes of Health, the Bill and Melinda Gates Foundation and the US Centers for Disease Control and Prevention.

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
