## [Reviewer comments · Proceedings of the Royal Society B: Biological Sciences]

Review History

RSPB-2019-0213.R0 (Original submission)

Review form: Reviewer 1

Recommendation

Major revision is needed (please make suggestions in comments)

Scientific importance: Is the manuscript an original and important contribution to its field?

Good

General interest: Is the paper of sufficient general interest?

Good

Quality of the paper: Is the overall quality of the paper suitable?

Marginal

Is the length of the paper justified?

Yes

Should the paper be seen by a specialist statistical reviewer?

No

Do you have any concerns about statistical analyses in this paper? If so, please specify them explicitly in your report.

No

It is a condition of publication that authors make their supporting data, code and materials available - either as supplementary material or hosted in an external repository. Please rate, if applicable, the supporting data on the following criteria.

Is it accessible?

Yes

Is it clear?

Yes

Is it adequate?

Yes

Do you have any ethical concerns with this paper?

No

Comments to the Author

This is a very interesting paper, and potentially has lots of powerful insights - but these are unfortunately hidden by a lack of detail. I appreciate that there are word limits to the publication, but much could be communicated to the reader without adding too much extra text.

For example:

1) The model is fitted using the methods of (28,29). This tells the reader very little. Is this fitting the deterministic or stochastic version of the model. What statistically characterises a "good fit". I was surprised that imports of infection are not mentioned - what stops the stochastic model from going extinct?

2) Similarly, it is stated that "We then estimated the coupling value for ...". How? This is a very difficult problem, especially if the coupling is relatively low, or the populations are closely synchronised. Much more detail is needed.

3) The different regions have different amplitudes and phases (described as "higher phases" on page 4 - a term I don't think informs). While the difference in amplitude has some mechanistic basis (poorer regions may be more impacted by seasonal weather), there is no mechanistic argument for the very different phases.

In fact I'm confused if the authors are arguing for birth rate leading to the dominant behaviour, or whether it is the seasonal forcing. These two factors are never disentangled.

Moreover, I cannot see any reason why the phase of the forcing would have any impact on the dominant periodicity (figure 3). Surely the phase simply shifts the outbreaks in time, but don't change their period?

4) Looking at the power spectra in the Supplement, it is clear that some regions show relatively poor agreement between models and data (inner South, outer West). It would be good to be more honest about this lack of fit in the main paper, and speculate as to the reason.

Minor Points

1) Crude Birth Rate needs to be defined.

2) Page 4 "two shape transmission parameters" should be "two dimensional transmission parameters"

2) Bottom of page 4. I don't understand the phrase "population mixing in between region synchrony"

3) Lots of the references have { } in the middle !

4) Figure 2, why is 1898 a larger epidemic year than all the others - some biological explanation is needed.

5) Figure 4 captions say "Note the different x-axis on the top row" - I think it would be clearer to

say "The top and bottom rows have different scales on the x axis".

6) The figures within the supplement do not tie-up with the text. For example the text refers to Figure S5A and S5B, whereas the figures are actually S8A and S8B

Review form: Reviewer 2

Recommendation

Accept with minor revision (please list in comments)

Scientific importance: Is the manuscript an original and important contribution to its field?

Excellent

General interest: Is the paper of sufficient general interest?

Good

Quality of the paper: Is the overall quality of the paper suitable?

Excellent

Is the length of the paper justified?

Yes

Should the paper be seen by a specialist statistical reviewer?

Yes

Do you have any concerns about statistical analyses in this paper? If so, please specify them explicitly in your report.

No

It is a condition of publication that authors make their supporting data, code and materials available - either as supplementary material or hosted in an external repository. Please rate, if applicable, the supporting data on the following criteria.

Is it accessible?

Yes

Is it clear?

Yes

Is it adequate?

Yes

Do you have any ethical concerns with this paper?

No

Comments to the Author

Specific points and typos

1. Both "test bed" and "testbed" are used in the text. Please use one only.
2. I suggest using only the term coupling parameter; "mixing" is occasionally used in the text as a synonym, but it is confusing.
3. Include a table of parameter estimates for all regions.
4. Line 22 Says "underling" instead of "underlying".
5. Line 47 What is meant by "density" of human mixing?
6. Line 55 Does "synchrony across local population" refer to insignificant phase differences

between city-level subdivisions?

7. Line 56-57 Is the period truly approximated by the dominant period of the largest subpopulation, or by the period of the whole population?
 8. Line 110 In what sense was the deterministic model simulated? Was it integrated numerically, for each parameter set? Please clarify and provide the details.
 9. Line 111 If this is the analysis shown in figure 3, please refer to figure. If results are not shown, please state this.
 10. Line 130 Please detail how coupling parameters were estimated and the way in which phase differences were quantified.
 11. Line 137 Starting year should be 1897. Maybe write "from 1897 to 1906", instead of using the dash?
 12. Line 159 How did the model capture historical CFR? Please explain whether and how it was estimated.
 13. Line 169 Similar to my comment to line 110, in what sense was the deterministic model simulated?
 14. Line 174 Should read: "one-, two- and greater than two-year".
 15. Line 179 Should read: "predictive".
 16. Line 181 Should read: "to gain insight".
 17. Line 182 Maybe omit the word "between" from the sentence?
 18. Line 242 Should read: "explains".
 19. Line 181 Omit comma after "analyzing".
 20. Figure 1 How well connected internally is each region? The South seems much larger, and probably less dense than the other inner regions. The Outer East is cut in two by the river, it seems odd to consider it an epidemiological subpopulation.
 21. Figure 3A The legend is unclear: I cannot tell West from Outer South or Central from Outer North. Gray and pale blue are also very hard to distinguish from black and blue, and I cannot find the dot-dashed key on the legend.
- Supplementary material
1. Draw a line in figures S1, S2, S4, S5 to indicate which values correspond to statistically significant power.
 2. Revise the expression "albeit slightly noisier". Maybe rephrase as "even if slightly noisier" or "albeit in a noisier fashion"?
 3. The fit of model time series to data, or to data behavior (periodicity) is not quantified. Confidence intervals for one and the other are shown as a visual test of the goodness of the fit.
 4. Line 57 "parameters matched well with the predicted dynamics, which showed strongly in phase dynamics, well."
 5. Line 92 It may be possible the that stochasticity,..."
 6. Figure S6 Please clarify whether the percentages shown are taken for each mixing rate, and not over all possible rates used.

Decision letter (RSPB-2019-0213.R0)

08-Apr-2019

Dear Mr Becker:

I am writing to inform you that your manuscript RSPB-2019-0213 entitled "Coexisting attractors in the context of cross-scale population dynamics: measles in London as a case study" has, in its current form, been rejected for publication in Proceedings B.

This action has been taken on the advice of referees, who have recommended that revisions are necessary. With this in mind we would be happy to consider a resubmission, provided the

comments of the referees are fully addressed. However please note that this is not a provisional acceptance.

In addition to the comments by the reviewers and the Associate Editor, I want to make a remark on John Brownlee. You write in lines 71-72 that John Brownlee "lacked the mechanistic framework now available to researchers". This is not entirely correct. Of course the field has advanced significantly, but the main reason why Brownlee could not do such analyses was a fundamental difference in how he approached the subject and how he felt epidemic phenomena should be explained. See a paper by Paul Fine (1979 'John Brownlee and the measurement of infectiousness ...', *J. Roy Stat Soc A*, 142, 347-362) and my own paper 'The law of mass action in epidemiology: a historical perspective' in Cuddington & Beisner (Eds.) 'Ecological paradigms lost: roots of theory change', 2005. Brownlee approached the topic a posteriori (by fitting curves to outbreaks that had occurred) and thought that reduction and gain of infective power by the infectious agent were the mechanism behind outbreaks. This in contrast to notably Ronald Ross, who worked a priori from basic assumptions, and thought that it was the availability of susceptibles that should be the mechanistic basis. Ross brought forward his approach clearly in 1911 and again in a series of publications from 1916. He and Brownlee had a polemic about the approaches and ultimately, Brownlee conceded that his approach and view of the matter was not the correct one. I'm not suggesting that you go into all this (or even refer to the above papers), but perhaps the wording can be slightly adapted. Brownlee certainly has a theory and considerations, they were just not the theory and considerations that survived in history to what we have today. That alle stemmed from Ross' approach developed concurrently with Brownlee's approach (and well-known to him). One can only imagine what insights Brownlee could have gained with his data had he adapted Ross' approach from the beginning.

Sincerely,
Hans Heesterbeek, Editor Proceedings B
mailto: proceedingsb@royalsociety.org

Associate Editor
Board Member: 1
Comments to Author:

This is very nice work, presenting a dataset which is unusual in breaking down measles dynamics within city pre-vaccination. There's a story with differing dynamic regimes between

regions of London (1- or 2- year periodic), yet the aggregate would appear as simply annual dynamics - this is surprising given that one would expect the dynamics to be so closely coupled, but the authors present a plausible explanation of this in a modelling framework.

Both referees raise points to address. Referee 1 raises some major points, particularly focusing on clarification of what was done. Referee 2 gives an extensive list of minor points for clarification also.

A few points from my own reading:

- surprised not to see comment on using mortality data: does this add any issues, e.g. seasonal bias over case data as CFR changes?
- the SI format of the word document mangles some of the equations. Maybe safer to submit it as PDF.
- as one of the referees mentions, could do with being more upfront about data limitations (e.g. in fig 1, outer South looks super noisy, and outer North is not at all convincingly annual)
- Fig S8 (two patch sims), there's background grid lines for years, but make the axes ticks also integer number of years
- Penultimate paragraph of main manuscript, can the differing birth rates (as shown in fig 1 region colours) be discounted as being the key difference between the regions driving the dynamics?

Reviewer(s)' Comments to Author:

Referee: 1

Comments to the Author(s)

This is a very interesting paper, and potentially has lots of powerful insights - but these are unfortunately hidden by a lack of detail. I appreciate that there are word limits to the publication, but much could be communicated to the reader without adding too much extra text.

For example:

- 1) The model is fitted using the methods of (28,29). This tells the reader very little. Is this fitting the deterministic or stochastic version of the model. What statistically characterises a "good fit". I was surprised that imports of infection are not mentioned - what stops the stochastic model from going extinct?
- 2) Similarly, it is stated that "We then estimated the coupling value for ...". How? This is a very difficult problem, especially if the coupling is relatively low, or the populations are closely synchronised. Much more detail is needed.
- 3) The different regions have different amplitudes and phases (described as "higher phases" on page 4 - a term I don't think informs). While the difference in amplitude has some mechanistic basis (poorer regions may be more impacted by seasonal weather), there is no mechanistic argument for the very different phases.

In fact I'm confused if the authors are arguing for birth rate leading to the dominant behaviour, or whether it is the seasonal forcing. These two factors are never disentangled.

Moreover, I cannot see any reason why the phase of the forcing would have any impact on the dominant periodicity (figure 3). Surely the phase simply shifts the outbreaks in time, but don't change their period?

- 4) Looking at the power spectra in the Supplement, it is clear that some regions show relatively poor agreement between models and data (inner South, outer West). It would be good to be more honest about this lack of fit in the main paper, and speculate as to the reason.

Minor Points

- 1) Crude Birth Rate needs to be defined.
- 2) Page 4 "two shape transmission parameters" should be "two dimensional transmission parameters"

- 2) Bottom of page 4. I don't understand the phrase "population mixing in between region synchrony"
- 3) Lots of the references have { } in the middle !
- 4) Figure 2, why is 1898 a larger epidemic year than all the others - some biological explanation is needed.
- 5) Figure 4 captions say "Note the different x-axis on the top row" - I think it would be clearer to say "The top and bottom rows have different scales on the x axis".
- 6) The figures within the supplement do not tie-up with the text. For example the text refers to Figure S5A and S5B, whereas the figures are actually S8A and S8B

Referee: 2

Comments to the Author(s)

Specific points and typos

1. Both "test bed" and "testbed" are used in the text. Please use one only.
2. I suggest using only the term coupling parameter; "mixing" is occasionally used in the text as a synonym, but it is confusing.
3. Include a table of parameter estimates for all regions.
4. Line 22 Says "underling" instead of "underlying".
5. Line 47 What is meant by "density" of human mixing?
6. Line 55 Does "synchrony across local population" refer to insignificant phase differences between city-level subdivisions?
7. Line 56-57 Is the period truly approximated by the dominant period of the largest subpopulation, or by the period of the whole population?
8. Line 110 In what sense was the deterministic model simulated? Was it integrated numerically, for each parameter set? Please clarify and provide the details.
9. Line 111 If this is the analysis shown in figure 3, please refer to figure. If results are not shown, please state this.
10. Line 130 Please detail how coupling parameters were estimated and the way in which phase differences were quantified.
11. Line 137 Starting year should be 1897. Maybe write "from 1897 to 1906", instead of using the dash?
12. Line 159 How did the model capture historical CFR? Please explain whether and how it was estimated.
13. Line 169 Similar to my comment to line 110, in what sense was the deterministic model simulated?
14. Line 174 Should read: "one-, two- and greater than two-year".
15. Line 179 Should read: "predictive".
16. Line 181 Should read: "to gain insight".
17. Line 182 Maybe omit the word "between" from the sentence?
18. Line 242 Should read: "explains".
19. Line 181 Omit comma after "analyzing".
20. Figure 1 How well connected internally is each region? The South seems much larger, and probably less dense than the other inner regions. The Outer East is cut in two by the river, it seems odd to consider it an epidemiological subpopulation.
21. Figure 3A The legend is unclear: I cannot tell West from Outer South or Central from Outer North. Gray and pale blue are also very hard to distinguish from black and blue, and I cannot find the dot-dashed key on the legend.

Supplementary material

1. Draw a line in figures S1, S2, S4, S5 to indicate which values correspond to statistically significant power.
2. Revise the expression "albeit slightly noisier". Maybe rephrase as "even if slightly noisier" or "albeit in a noisier fashion"?
3. The fit of model time series to data, or to data behavior (periodicity) is not quantified. Confidence intervals for one and the other are shown as a visual test of the goodness of the fit.

4. Line 57 “parameters matched well with the predicted dynamics, which showed strongly in phase dynamics, well.”
5. Line 92 It may be possible the that stochasticity,...”
6. Figure S6 Please clarify whether the percentages shown are taken for each mixing rate, and not over all possible rates used.

Author's Response to Decision Letter for (RSPB-2019-0213.R0)

See Appendix A.

RSPB-2019-1510.R0

Review form: Reviewer 2

Recommendation

Accept with minor revision (please list in comments)

Scientific importance: Is the manuscript an original and important contribution to its field?

Good

General interest: Is the paper of sufficient general interest?

Good

Quality of the paper: Is the overall quality of the paper suitable?

Good

Is the length of the paper justified?

Yes

Should the paper be seen by a specialist statistical reviewer?

Yes

Do you have any concerns about statistical analyses in this paper? If so, please specify them explicitly in your report.

Yes

It is a condition of publication that authors make their supporting data, code and materials available - either as supplementary material or hosted in an external repository. Please rate, if applicable, the supporting data on the following criteria.

Is it accessible?

Yes

Is it clear?

Yes

Is it adequate?

Yes

Do you have any ethical concerns with this paper?

No

Comments to the Author

Thank you for the details on the fitting procedure. I would appreciate if the authors could distinguish exactly what parameters are fitted at each stage: are monthly transmission rates estimated alongside CFR, importation rate and "measurement" error, with the stochastic SEIR? How is the sinusoidal transmission function fitted?

I presume monthly transmission rates are fitted with the stochastic SEIR (though it seems like a lot of parameters are fitted at once), and those estimates are then fitted using cubic splines. What exactly are the coefficients b_i given in the table? I am also unsure of how the amplitude and phase of the sinusoidal transmission function are estimated from the monthly rates; monthly estimates appear to have more than one maximum for the biennial regime, which would suggest the sinusoidal function may need a modification in frequency too - unless the authors are picking the phase and amplitude that lead to epidemic cycles of the observed periodicity.

The authors explain that the deterministic SEIR is integrated numerically in order to obtain figure 3; I recommend referring to this figure in that part of the text.

I thank the authors for including a table of parameter and initial condition estimates. Is it possible to include an explanatory text for the table? I would suggest leaving phase and amplitude estimates to another table, or marking in some way that they are the result of another round of fitting.

The phase of two different things may be relevant to this work, but they are not always clearly distinguished: one is the phase of the sinusoidal forcing function; the other is the epidemic phase of a given time series, and the paper does not explain whether it is quantified and how. Moreover, I am not convinced of using epidemic phase to discuss epidemic synchrony between populations: I understood that the authors seek to demonstrate that different dynamic regimes (e.g., annual and biennial epidemic cycles) can coexist in the presence of intermediate or high coupling rates, not whether outbreaks are in phase. If epidemic phase is indeed what they want to compare, please explain the methodology used to do so.

I would prefer to see the confidence levels shown in the response in the paper; to me, showing them does not subtract from the results. Along those lines, I suggest commenting on the coupling rate estimates: according to figure S7, only two or three pairs have a coupling rate significantly different from 0.

Four parameters receive duplicated labels (and I wasted some time realising this!): ϕ (CFR and phase) and σ (incubation and coupling rate). Could you please give them different labels?

I was a little surprised to see such low estimates of case fatality rate for the era. Presumably, measles case fatality has declined over time.

The lower graphs in figure 4 and S6 give the percent of simulations that are out of phase for various coupling rates. Please state that this percentage is taken over all simulations with that coupling rate (this may be obvious in the figure of the main text, but is not on the one that is in the supplement).

Main text line number:

74] Maybe rephrase along the lines of: "utilized (24, 25) and necessary to estimate seasonal transmission parameters and test the role of demographic changes on epidemic dynamics."

86] and an approximate four outer

109] Should read "forward simulated"?

110] Refer also to the supplement for additional details on the fitting process.

115] How are the values for phase and amplitude selected? It seemed like the transmission term of the fitted model was not sinusoidal.
 189] “these two dimensional transmission parameters”

Supplement

Model description is now slightly mangled: it is mentioned that a stochastic version of the equations is used, but the equations include stochastic terms.

Please give more details of the fitting procedure that yields cubic splines for the seasonal transmission rate and, if adequate, list the coefficients for the cubic splines (ξ_i) that appear in its definition. Are the authors first finding estimates for the monthly transmission rate, and fitting these set of estimates with cubic splines?

How well do model dynamics correlate when changing from the estimated monthly transmission rates to sinusoidal forcing?

It is unclear which parameter values come from which fitting process, since both the coefficients b_i and the phase are given in the table, but I had understood they came from different stages of the fitting.

Estimates for R_0 seem extremely high, particularly for the Central and East regions, how do you interpret this?

6] Model description is now slightly mangled: it is mentioned that a stochastic version of the equations is used, but the equations include stochastic terms.

22] I am confused by the equation used to fit β_t , which I took to be the transmission rate in biweek t , and cannot see why a single number is fitted using cubic splines. Presumably, the splines really fit the annual variation in transmission (is this the case?), but then I would have expected to see some explicit dependence on time of the expression defining β_t . I take it the ξ_i depend on t . Is this correct? How does this connect to the sinusoidal transmission term mentioned in the main text?

30] State upfront that underreporting could be estimated as part of the procedure, but it will be assumed that reporting is perfect. The phrasing is confusing at present. Also, since the expression for the parameters of the cumulative normal distribution is simplified by this assumption, it may be wise to write it in the equations of lines 33 and 38.

33] Any reason why the value 0.5 is preferred?

41] Are parameters allowed to change simultaneously? (What is the dimension of the space in which these random walks take place?)

44] How do 60 iterations render 400 parameters? (This is probably not what the authors are saying, could they please clarify?)

45] Am I correct in thinking that you have a set of parameter sets?

54] “due to aggregation of” ... Disconnected populations?

120] for each simulation set of parameters... to produce a set of theoretical predictions.

122] Do you mean that you “fit a sinusoidal contact rate to the inferred monthly transmission rate and obtain estimates of amplitude and phase”?

Review form: Reviewer 3

Recommendation

Major revision is needed (please make suggestions in comments)

Scientific importance: Is the manuscript an original and important contribution to its field?
Good

General interest: Is the paper of sufficient general interest?
Excellent

Quality of the paper: Is the overall quality of the paper suitable?
Marginal

Is the length of the paper justified?
Yes

Should the paper be seen by a specialist statistical reviewer?
No

Do you have any concerns about statistical analyses in this paper? If so, please specify them explicitly in your report.
Yes

It is a condition of publication that authors make their supporting data, code and materials available - either as supplementary material or hosted in an external repository. Please rate, if applicable, the supporting data on the following criteria.

Is it accessible?
Yes

Is it clear?
Yes

Is it adequate?
Yes

Do you have any ethical concerns with this paper?
No

Comments to the Author

This manuscript presents data and modeling analyses showing a surprising diversity of measles dynamics for different regions within the city of London, during a decade in the prevaccination period. Despite patterns that appear strongly annual at the aggregate city-wide level, the authors show that dynamics within each of nine sub-regions divide between annual, biennial, and a few that have less distinct period. This empirical finding is an interesting insight into city-scale measles dynamics, which as the authors note have been a canonical success story of simple mass-action models, which assume random mixing and average over any internal heterogeneities. This is inherently interesting and has potential to advance our understanding of basic principles of disease dynamics. The authors analyze the data using a series of modeling and statistical approaches. Their major findings center on the role of seasonality in transmission in driving the observed differences in periodicity among sub-regions, and on the influence of coupling between sub-regions on synchrony/asynchrony. They conclude that they can explain the diversity of dynamics on theoretical grounds, and they draw parallels to other recent studies of measles or theoretical disease dynamics.

I am joining this review process after a previous round of review and resubmission was completed. I read and considered the ms in its entirety, and formed my opinion, then read the response to reviewers to see if it shed any light on my outstanding questions and concerns. Below I will summarize these, making minimal reference to the previous reviews and responses.

On the whole I was very intrigued by the dataset, and agreed with the broad outline of the authors' approach to analyzing it. The conclusions seemed plausible, at a qualitative level, and the framing in the introduction and discussion was excellent. However there were numerous issues in the methods and results that raised concerns for me, including many issues of clarity/communication that I was surprised to see in a resubmitted manuscript. As a result I could not quite get on board with the repeated claims of strong agreement between theory and data. I summarize my concerns below.

1. A basic description of the data source was missing, and the brief mentions throughout the ms were slightly confusing. The paper is initially framed in terms of "borough mortality records" (line 66) but it isn't explained how these relate to measles cases. Is measles listed as the cause of death? Is this unambiguous, always? Line 86 mentions "measles notifications", which sounds like a different data source from borough mortality records. The caption to Figure 1A says "monthly measles mortality reports", while 1B says just "Mortality". I am guessing all of these refer to the same data source, but am uncertain, and a bit unclear what that data source actually is.

2. The primary result of the modeling analysis is the conclusion that local variation in seasonality drives the variation in periodicity among regions. The chief finding from the data analysis is shown in Fig 3A, which superimposes the inferred seasonal transmission curves for all regions. Then these findings are related to the theoretical predictions in the bifurcation diagram in Fig 3B, which shows how different amplitude and phase parameters gave rise to different periodicity in a deterministic simulation model. The broad contours of this result are interesting, but I was left with lots of questions. The most important of these was how the seasonality models used in the two analyses relate to each other. Fig 3A uses a spline-based model, which leads to the (curious) double-peaked functions shown in the figure. But the analysis in Fig 3B uses a sinusoidal curve with period 1 year. How well can this function capture the empirical patterns from the data? The text (lines 121-123) acknowledges this was an approximate fit, but the fit is never shown, and no uncertainties are given for the fitted amplitude and phase parameters. Particularly for the 2-year group (shown in blue), where the inferred seasonality looks almost like a sinusoid with a period of 0.5 year, I can't really imagine how a period-1 sinusoid could fit it. This undermined my faith in this core result of the ms.

There were also more minor questions about this figure set, including:

- In Fig 3A, which region is in which periodicity group? Are any regions in the 2+ group? The black and gray colors are difficult to tell apart in the figure.
- why are there 13 months time-points for each series in Fig 3A? What is the meaning of the extra point after December?
- Fig 3B shows ten separate points, representing the parameters of each region plus the aggregate city (which should be omitted, or shown with a different symbol). These are impossible to evaluate without seeing the inferred seasonality traces for each region. Indeed I had originally noted that Fig 3A should 'connect the dots' to show individual traces, but I gather from the response to reviewers that the authors deliberately removed these.
- It is never stated how exactly the periodicity is assessed for each time series (in the data or the deterministic simulations). From the power spectra shown in Figs S1 and S2, the period is sometimes ambiguous (e.g. outer South, outer West).

3. The second major result from the modeling analyses addressed the influence of coupling between regions. The outputs of this analysis were posed entirely in terms of whether the dynamics of two patches were in-phase or out-of-phase. I do not understand what this means in the context of the North-West pair. How can an annual signal (North) be out of phase with a biennial signal (West)? Similarly, on pp6-7 of the SI, there are strong statements about East-Central and East-Outer East all being strongly in-phase. What does this mean, since they're all annual? At the bottom line, I felt like I was unable to understand how the authors were talking about in-phase/out-of-phase throughout the manuscript. How were these terms defined, and how was the determination made for the quantitative analyses shown in Figs 4 and S6?

4. The section on coupling rates made no mention of the importation rate, $iota$, estimated in the single-region analyses. There is a statement (lines 138-139) that local parameters were “inherited from the regional analysis” for the coupling analysis – did this include the importation rate? Also, the parameter estimates shown in the SI table show a large range of $iota$ values estimates for the regions (from 4.71 to 80.43). How did these values relate to the pairwise coupling values estimated in Fig S7?

5. Throughout the ms, I was struck by the many claims of ‘generally close fits’ and ‘strong visual fit’ and so on. This struck me as strong language which was not always borne out by the figures, e.g. Figs S3, S4 and S5 are not obviously “close fits”. Even Fig 2 is debatable, since the model does a good job with the big 1898 epidemic, the two-year cycle in the West region, and the general envelope of amplitudes, but otherwise there’s not much sign that the fitted model curves are correlated with the data. I noted the discussion of this point in the response document, and I’m sympathetic to the authors’ position. However I think their case would be stronger if they addressed the specific elements of the model fit that they believe support their case. I found the current phrasing (strong but vague claims of close fit) to distract and slightly undermine the claims that the theory explained the data.

6. There are numerous small technical errors throughout the manuscript. Some of these have limited bearing on the take-home conclusions, but they don’t inspire confidence (in particular it makes me worry about the engagement of the senior authors, who certainly understand these issues). Other issues are potentially more troubling. For instance:

- Line 118, allowing the phase parameters to take values from 0-2 doesn’t correspond to a maximum 2-month lag. The expression is in units of radians, so it is $2/2*\pi$ years, or roughly 4 months.
- Lines 132-134, $\sigma=0.01$ does not mean that the infected population in Y contributes 1% of infections in population X. This depends on the absolute values of I_Y and I_X . The σ gives a relative weighting to those contributions, which will scale with current prevalence on each patch.
- Lines 143-144 and 215-216, $\sigma = 1$ does not correspond to a well-mixed system. This would be $\sigma=0.5$. $\sigma=1$ is a curious case where only I_Y contribute to infection on patch X. As a consequence, I think Fig 4CD should be restricted to the range $\sigma = 0$ to 0.5.
- Line 130, which N is used here? In the original Keeling & Rohani paper all patches had equal size, but here it matters.
- As presented and defined in the SI, the modeling of the importation rate is wrong. This process, represented by $iota$, is defined as the ‘yearly infected importation rate’. This gives it units of cases over time. But in the model on page 1 of the SI, it must have units of cases as it is added to I (the number of cases) and multiplied by λ to contribute to the incidence of new infections.
- Even if this is treated as a minor semantic issue, it raises quantitative concerns since the table of parameter estimates on SI page 4 reports values up to $iota = 80$ for individual regions. If this value is used in the equations, as written, it would be a major driver of the dynamics. If it is modified somehow, this needs to be explained. There is a vague statement about the stochastic version of the model, based on Euler transitions (a term I’m not familiar with. Is this equivalent to the Gillespie algorithm? If not, how is it different?). Regardless, this does not clarify how $iota$ is treated since it has equivalent status to the state variable I.
- Line 103, what is a ‘dispersed normal distribution’? Reading the relevant part of the SI, should this be ‘discretized’?
- The description of the observation model (SI lines 28-39) is impossible to understand without reference to the cited source. There is an error in the first equation (final term lacks proper exponents) and the notation is sloppy (use of comma for semicolon, so I was initially confused about how the normal distribution could have 3 parameters). In lines 30-31, why is ρ set equal to 1, and what is ‘measurement error’ given that under-reporting is included separately? Is it for false positives? In line 36, does ‘mortality’ refer to true deaths or observed deaths?
- SI line 22, this is insufficient detail for me to understand how the cubic spline fitting was done.

Minor points

- Several symbols are used for multiple quantities in the ms. ϕ is used for both the case fatality rate (SI page 1, table on page 4) and the phase of the sinusoidal transmission function (SI page 8, Fig 3)? σ is both the inverse of latent period and the coupling rate.
- I found it strange that results for the aggregate city-wide data were mixed in with the results for the regions. In figs 3A and 3B, this would be easily remedied by using a different symbol for the aggregate data. At other points, e.g. on lines 116-117 where the mean transmission rate used for simulations seems to be an average of the aggregate parameter estimate averaged with the regional estimates, I would suggest it should be removed.
- Line 99 – relative to what? Why then is it appropriate to use the quantitative estimates to claim support for the theory (lines 176-179), without caveat?
- Very little information given about the iterated filtering. How was convergence assessed? How many runs were done?
- Line 151 – it would be helpful to give all population sizes somewhere.
- Line 158 – what is the basis for this statement about susceptible population sizes? It seems to hinge on attack rates, ages of infection, etc, and I could not work all this out in my head.
- Line 175 – I couldn't understand what this ('indicating potential periodic variation' was trying to say).
- Line 210 – I was puzzled by the statement about differences in phase modulating the effective R_0 at the start of the year. I have since read the authors' explanation in their response to reviewers, which is more information but still not very convincing to me. It seems to me that this phase effect differs importantly from the effect of population susceptibility, because it is transient and will average out over the course of the season, whereas the effect of susceptibility applies to the whole season.
- Line 260 – in what way are the findings consistent with Keeling & Rohani's theory, which predicts that synchrony rises with coupling up to a critical value, then declines beyond that. There is perhaps a hint of this in Fig 4C?
- Line 293 – is this statement consistent with the earlier discussion about the decline of synchrony beyond a critical coupling level?

Decision letter (RSPB-2019-1510.R0)

19-Sep-2019

Dear Mr Becker:

Your manuscript has now been peer reviewed and the reviews have been assessed by an Associate Editor. The reviewers' comments (not including confidential comments to the Editor) and the comments from the Associate Editor are included at the end of this email for your reference. As you will see, the reviewers have raised important concerns with your manuscript and we would like to invite you to revise your manuscript to address them. One of the reviewers (referee 3 below) is new to the manuscript and it is fitting that you should get a chance to respond. However, given the nature of the detailed and critical comments by this reviewer it is good to realize that the manuscript is borderline between allowing a further revision and rejection, as we usually require much more convergence, after a round of major revision you just went through, in the level at which reviewers are convinced by the findings.

Research ethics:

Use of animals and field studies:

All supplementary materials accompanying an accepted article will be treated as in their final form. They will be published alongside the paper on the journal website and posted on the online

figshare repository. Files on figshare will be made available approximately one week before the accompanying article so that the supplementary material can be attributed a unique DOI. Please try to submit all supplementary material as a single file.

Please submit a copy of your revised paper within three weeks. If we do not hear from you within this time your manuscript will be rejected. If you are unable to meet this deadline please let us know as soon as possible, as we may be able to grant a short extension.

Best wishes,
Professor Hans Heesterbeek
mailto:proceedingsb@royalsociety.org

Reviewer(s)' Comments to Author:

Referee: 2

Comments to the Author(s).

Thank you for the details on the fitting procedure. I would appreciate if the authors could distinguish exactly what parameters are fitted at each stage: are monthly transmission rates estimated alongside CFR, importation rate and "measurement" error, with the stochastic SEIR? How is the sinusoidal transmission function fitted?

I presume monthly transmission rates are fitted with the stochastic SEIR (though it seems like a lot of parameters are fitted at once), and those estimates are then fitted using cubic splines. What exactly are the coefficients b_i given in the table? I am also unsure of how the amplitude and phase of the sinusoidal transmission function are estimated from the monthly rates; monthly estimates appear to have more than one maximum for the biennial regime, which would suggest the sinusoidal function may need a modification in frequency too - unless the authors are picking the phase and amplitude that lead to epidemic cycles of the observed periodicity.

The authors explain that the deterministic SEIR is integrated numerically in order to obtain figure 3; I recommend referring to this figure in that part of the text.

I thank the authors for including a table of parameter and initial condition estimates. Is it possible to include an explanatory text for the table? I would suggest leaving phase and amplitude estimates to another table, or marking in some way that they are the result of another round of fitting.

The phase of two different things may be relevant to this work, but they are not always clearly distinguished: one is the phase of the sinusoidal forcing function; the other is the epidemic phase of a given time series, and the paper does not explain whether it is quantified and how. Moreover, I am not convinced of using epidemic phase to discuss epidemic synchrony between populations: I understood that the authors seek to demonstrate that different dynamic regimes (e.g., annual and biennial epidemic cycles) can coexist in the presence of intermediate or high coupling rates, not whether outbreaks are in phase. If epidemic phase is indeed what they want to compare, please explain the methodology used to do so.

I would prefer to see the confidence levels shown in the response in the paper; to me, showing them does not subtract from the results. Along those lines, I suggest commenting on the coupling rate estimates: according to figure S7, only two or three pairs have a coupling rate significantly different from 0.

Four parameters receive duplicated labels (and I wasted some time realising this!): ϕ (CFR and phase) and σ (incubation and coupling rate). Could you please give them different labels?

I was a little surprised to see such low estimates of case fatality rate for the era. Presumably, measles case fatality has declined over time.

The lower graphs in figure 4 and S6 give the percent of simulations that are out of phase for various coupling rates. Please state that this percentage is taken over all simulations with that coupling rate (this may be obvious in the figure of the main text, but is not on the one that is in the supplement).

Main text line number:

74] Maybe rephrase along the lines of: "utilized (24, 25) and necessary to estimate seasonal transmission parameters and test the role of demographic changes on epidemic dynamics."

86] and an approximate four outer

109] Should read "forward simulated"?

110] Refer also to the supplement for additional details on the fitting process.

115] How are the values for phase and amplitude selected? It seemed like the transmission term of the fitted model was not sinusoidal.

189] "these two dimensional transmission parameters"

Supplement

Model description is now slightly mangled: it is mentioned that a stochastic version of the equations is used, but the equations include stochastic terms.

Please give more details of the fitting procedure that yields cubic splines for the seasonal transmission rate and, if adequate, list the coefficients for the cubic splines (ξ) that appear in its definition. Are the authors first finding estimates for the monthly transmission rate, and fitting these set of estimates with cubic splines?

How well do model dynamics correlate when changing from the estimated monthly transmission rates to sinusoidal forcing?

It is unclear which parameter values come from which fitting process, since both the coefficients b_i and the phase are given in the table, but I had understood they came from different stages of the fitting.

Estimates for R_0 seem extremely high, particularly for the Central and East regions, how do you interpret this?

6] Model description is now slightly mangled: it is mentioned that a stochastic version of the equations is used, but the equations include stochastic terms.

22] I am confused by the equation used to fit βt , which I took to be the transmission rate in biweek t , and cannot see why a single number is fitted using cubic splines. Presumably, the splines really fit the annual variation in transmission (is this the case?), but then I would have expected to see some explicit dependence on time of the expression defining βt . I take it the ξ

depend on t . Is this correct? How does this connect to the sinusoidal transmission term mentioned in the main text?

30] State upfront that underreporting could be estimated as part of the procedure, but it will be assumed that reporting is perfect. The phrasing is confusing at present. Also, since the expression for the parameters of the cumulative normal distribution is simplified by this assumption, it may be wise to write it in the equations of lines 33 and 38.

33] Any reason why the value 0.5 is preferred?

41] Are parameters allowed to change simultaneously? (What is the dimension of the space in which these random walks take place?)

44] How do 60 iterations render 400 parameters? (This is probably not what the authors are saying, could they please clarify?)

45] Am I correct in thinking that you have a set of parameter sets?

54] "due to aggregation of" ... Disconnected populations?

120] for each simulation set of parameters... to produce a set of theoretical predictions.

122] Do you mean that you "fit a sinusoidal contact rate to the inferred monthly transmission rate and obtain estimates of amplitude and phase"?

Referee: 3

Comments to the Author(s).

This manuscript presents data and modeling analyses showing a surprising diversity of measles dynamics for different regions within the city of London, during a decade in the prevaccination period. Despite patterns that appear strongly annual at the aggregate city-wide level, the authors show that dynamics within each of nine sub-regions divide between annual, biennial, and a few that have less distinct period. This empirical finding is an interesting insight into city-scale measles dynamics, which as the authors note have been a canonical success story of simple mass-action models, which assume random mixing and average over any internal heterogeneities. This is inherently interesting and has potential to advance our understanding of basic principles of disease dynamics. The authors analyze the data using a series of modeling and statistical approaches. Their major findings center on the role of seasonality in transmission in driving the observed differences in periodicity among sub-regions, and on the influence of coupling between sub-regions on synchrony/asynchrony. They conclude that they can explain the diversity of dynamics on theoretical grounds, and they draw parallels to other recent studies of measles or theoretical disease dynamics.

I am joining this review process after a previous round of review and resubmission was completed. I read and considered the ms in its entirety, and formed my opinion, then read the response to reviewers to see if it shed any light on my outstanding questions and concerns. Below I will summarize these, making minimal reference to the previous reviews and responses.

On the whole I was very intrigued by the dataset, and agreed with the broad outline of the authors' approach to analyzing it. The conclusions seemed plausible, at a qualitative level, and the framing in the introduction and discussion was excellent. However there were numerous issues in the methods and results that raised concerns for me, including many issues of clarity/communication that I was surprised to see in a resubmitted manuscript. As a result I could not quite get on board with the repeated claims of strong agreement between theory and data. I summarize my concerns below.

1. A basic description of the data source was missing, and the brief mentions throughout the ms were slightly confusing. The paper is initially framed in terms of "borough mortality records" (line 66) but it isn't explained how these relate to measles cases. Is measles listed as the cause of death? Is this unambiguous, always? Line 86 mentions "measles notifications", which sounds like a different data source from borough mortality records. The caption to Figure 1A says "monthly measles mortality reports", while 1B says just "Mortality". I am guessing all of these

refer to the same data source, but am uncertain, and a bit unclear what that data source actually is.

2. The primary result of the modeling analysis is the conclusion that local variation in seasonality drives the variation in periodicity among regions. The chief finding from the data analysis is shown in Fig 3A, which superimposes the inferred seasonal transmission curves for all regions. Then these findings are related to the theoretical predictions in the bifurcation diagram in Fig 3B, which shows how different amplitude and phase parameters gave rise to different periodicity in a deterministic simulation model. The broad contours of this result are interesting, but I was left with lots of questions. The most important of these was how the seasonality models used in the two analyses relate to each other. Fig 3A uses a spline-based model, which leads to the (curious) double-peaked functions shown in the figure. But the analysis in Fig 3B uses a sinusoidal curve with period 1 year. How well can this function capture the empirical patterns from the data? The text (lines 121-123) acknowledges this was an approximate fit, but the fit is never shown, and no uncertainties are given for the fitted amplitude and phase parameters. Particularly for the 2-year group (shown in blue), where the inferred seasonality looks almost like a sinusoid with a period of 0.5 year, I can't really imagine how a period-1 sinusoid could fit it. This undermined my faith in this core result of the ms.

There were also more minor questions about this figure set, including:

- In Fig 3A, which region is in which periodicity group? Are any regions in the 2+ group? The black and gray colors are difficult to tell apart in the figure.
- why are there 13 months time-points for each series in Fig 3A? What is the meaning of the extra point after December?
- Fig 3B shows ten separate points, representing the parameters of each region plus the aggregate city (which should be omitted, or shown with a different symbol). These are impossible to evaluate without seeing the inferred seasonality traces for each region. Indeed I had originally noted that Fig 3A should 'connect the dots' to show individual traces, but I gather from the response to reviewers that the authors deliberately removed these.
- It is never stated how exactly the periodicity is assessed for each time series (in the data or the deterministic simulations). From the power spectra shown in Figs S1 and S2, the period is sometimes ambiguous (e.g. outer South, outer West).

3. The second major result from the modeling analyses addressed the influence of coupling between regions. The outputs of this analysis were posed entirely in terms of whether the dynamics of two patches were in-phase or out-of-phase. I do not understand what this means in the context of the North-West pair. How can an annual signal (North) be out of phase with a biennial signal (West)? Similarly, on pp6-7 of the SI, there are strong statements about East-Central and East-Outer East all being strongly in-phase. What does this mean, since they're all annual? At the bottom line, I felt like I was unable to understand how the authors were talking about in-phase/out-of-phase throughout the manuscript. How were these terms defined, and how was the determination made for the quantitative analyses shown in Figs 4 and S6?

4. The section on coupling rates made no mention of the importation rate, ι , estimated in the single-region analyses. There is a statement (lines 138-139) that local parameters were "inherited from the regional analysis" for the coupling analysis – did this include the importation rate? Also, the parameter estimates shown in the SI table show a large range of ι values estimates for the regions (from 4.71 to 80.43). How did these values relate to the pairwise coupling values estimated in Fig S7?

5. Throughout the ms, I was struck by the many claims of 'generally close fits' and 'strong visual fit' and so on. This struck me as strong language which was not always borne out by the figures, e.g. Figs S3, S4 and S5 are not obviously "close fits". Even Fig 2 is debatable, since the model does a good job with the big 1898 epidemic, the two-year cycle in the West region, and the general envelope of amplitudes, but otherwise there's not much sign that the fitted model curves are correlated with the data. I noted the discussion of this point in the response document, and

I'm sympathetic to the authors' position. However I think their case would be stronger if they addressed the specific elements of the model fit that they believe support their case. I found the current phrasing (strong but vague claims of close fit) to distract and slightly undermine the claims that the theory explained the data.

6. There are numerous small technical errors throughout the manuscript. Some of these have limited bearing on the take-home conclusions, but they don't inspire confidence (in particular it makes me worry about the engagement of the senior authors, who certainly understand these issues). Other issues are potentially more troubling. For instance:

- Line 118, allowing the phase parameters to take values from 0-2 doesn't correspond to a maximum 2-month lag. The expression is in units of radians, so it is $2/2\pi$ years, or roughly 4 months.

- Lines 132-134, $\sigma=0.01$ does not mean that the infected population in Y contributes 1% of infections in population X. This depends on the absolute values of I_Y and I_X . The sigma gives a relative weighting to those contributions, which will scale with current prevalence on each patch.

- Lines 143-144 and 215-216, $\sigma = 1$ does not correspond to a well-mixed system. This would be $\sigma=0.5$. $\sigma=1$ is a curious case where only I_Y contribute to infection on patch X. As a consequence, I think Fig 4CD should be restricted to the range $\sigma = 0$ to 0.5.

- Line 130, which N is used here? In the original Keeling & Rohani paper all patches had equal size, but here it matters.

- As presented and defined in the SI, the modeling of the importation rate is wrong. This process, represented by ι , is defined as the 'yearly infected importation rate'. This gives it units of cases over time. But in the model on page 1 of the SI, it must have units of cases as it is added to I (the number of cases) and multiplied by λ to contribute to the incidence of new infections.

- Even if this is treated as a minor semantic issue, it raises quantitative concerns since the table of parameter estimates on SI page 4 reports values up to $\iota = 80$ for individual regions. If this value is used in the equations, as written, it would be a major driver of the dynamics. If it is modified somehow, this needs to be explained. There is a vague statement about the stochastic version of the model, based on Euler transitions (a term I'm not familiar with. Is this equivalent to the Gillespie algorithm? If not, how is it different?). Regardless, this does not clarify how ι is treated since it has equivalent status to the state variable I.

- Line 103, what is a 'dispersed normal distribution'? Reading the relevant part of the SI, should this be 'discretized'?

- The description of the observation model (SI lines 28-39) is impossible to understand without reference to the cited source. There is an error in the first equation (final term lacks proper exponents) and the notation is sloppy (use of comma for semicolon, so I was initially confused about how the normal distribution could have 3 parameters). In lines 30-31, why is ρ set equal to 1, and what is 'measurement error' given that under-reporting is included separately? Is it for false positives? In line 36, does 'mortality' refer to true deaths or observed deaths?

- SI line 22, this is insufficient detail for me to understand how the cubic spline fitting was done.

Minor points

- Several symbols are used for multiple quantities in the ms. ϕ is used for both the case fatality rate (SI page 1, table on page 4) and the phase of the sinusoidal transmission function (SI page 8, Fig 3)? σ is both the inverse of latent period and the coupling rate.

- I found it strange that results for the aggregate city-wide data were mixed in with the results for the regions. In figs 3A and 3B, this would be easily remedied by using a different symbol for the aggregate data. At other points, e.g. on lines 116-117 where the mean transmission rate used for simulations seems to be an average of the aggregate parameter estimate averaged with the regional estimates, I would suggest it should be removed.

- Line 99 - relative to what? Why then is it appropriate to use the quantitative estimates to claim support for the theory (lines 176-179), without caveat?

- Very little information given about the iterated filtering. How was convergence assessed? How many runs were done?

- Line 151 - it would be helpful to give all population sizes somewhere.

- Line 158 – what is the basis for this statement about susceptible population sizes? It seems to hinge on attack rates, ages of infection, etc, and I could not work all this out in my head.
- Line 175 – I couldn't understand what this ('indicating potential periodic variation' was trying to say).
- Line 210 – I was puzzled by the statement about differences in phase modulating the effective R_0 at the start of the year. I have since read the authors' explanation in their response to reviewers, which is more information but still not very convincing to me. It seems to me that this phase effect differs importantly from the effect of population susceptibility, because it is transient and will average out over the course of the season, whereas the effect of susceptibility applies to the whole season.
- Line 260 – in what way are the findings consistent with Keeling & Rohani's theory, which predicts that synchrony rises with coupling up to a critical value, then declines beyond that. There is perhaps a hint of this in Fig 4C?
- Line 293 – is this statement consistent with the earlier discussion about the decline of synchrony beyond a critical coupling level?

Author's Response to Decision Letter for (RSPB-2019-1510.R0)

See Appendix B.

RSPB-2019-1510.R1 (Revision)

Review form: Reviewer 2

Recommendation

Accept with minor revision (please list in comments)

Scientific importance: Is the manuscript an original and important contribution to its field?

Good

General interest: Is the paper of sufficient general interest?

Acceptable

Quality of the paper: Is the overall quality of the paper suitable?

Good

Is the length of the paper justified?

Yes

Should the paper be seen by a specialist statistical reviewer?

Yes

Do you have any concerns about statistical analyses in this paper? If so, please specify them explicitly in your report.

No

It is a condition of publication that authors make their supporting data, code and materials available - either as supplementary material or hosted in an external repository. Please rate, if applicable, the supporting data on the following criteria.

Is it accessible?

Yes

Is it clear?

Yes

Is it adequate?

Yes

Do you have any ethical concerns with this paper?

No

Comments to the Author

The authors document measles time series of diverse dominant periodicity for London regions, estimate important model parameters from the data with the aid of demographic information, and reproduce to some extent the observed patterns by incorporating specific coupling rates and sinusoidal phase and amplitude transmission estimates into the model. The data set analysed is rather unique and their findings interesting to the field, but I strongly discourage the use of the term “synchrony” to refer to a coincidence in interepidemic period, simply because they are not interchangeable: biennial time series, for example, may present outbreaks in alternate years and thus be in opposite phases, a situation we would like to distinguish from biennial epidemics that peak simultaneously. Moreover, the authors later make reference to previous work that examined changes in synchrony among boroughs and cities as measured by the correlation between time series, further demonstrating that calling different notions the same leads to confusion.

Another aspect that could be discussed further is why is the importation rate assumed to be constant, when a fraction of the number of infectious individuals in the other patch at each time step could be used instead. Related to that, the text in [146] states “we used the seasonal transmission values” which is not the number of infectious individuals in the other population and I therefore do not find justified.

It does not make much sense to me that forcing phase plays the same role as initial conditions, could the authors provide more detail to support this argument?

Minor points:

[28] What is meant by “phase coupling”?

[89] The statement “the outer ring of London was not formally divided into regions” makes me wonder whether the mortality records used in this study were reported in the groupings used in this paper, or whether they were grouped in these regions by the authors. Which was the case?

[95] What resolution in time is assumed for the interpolated data? How are the interpolated data employed?

[114] The authors do clarify in their response to referees that they performed 60 simulations for each of the 400 sets of parameters used, but this not as clearly stated in the paper.

[123] Is the transmission rate “based on” or equal to the mean of their estimates?

[138] Could you provide more details as to how was regional isolation “relative” in the previous model? Were modelled populations not completely independent?

[182] How is population size related to interepidemic period (other than by defining whether the population is above the CCS)?

[196] I am unconvinced of the source used for CFR estimates. The cited paper (27) uses data from 1980 onward, while the paper under review analyses a data set from 1897-1906. Why would the CFR be similar in such different eras? It has definitely gone down since the turn of the 20th century, and in the 1980s plenty of treatments and medical knowledge were available that were not over eighty years earlier.

[198] Please discuss how incomplete reporting would modify these estimates.

[234] Please include references to the birth rates of each region in your discussion.

[277] What is meant by a “bifurcation grid”?

[304] I would expect lower school attendance to lower forcing amplitude, rather than modifying its phase.

Decision letter (RSPB-2019-1510.R1)

16-Dec-2019

Dear Mr Becker:

Your manuscript has now been peer reviewed and the review has been assessed by an Associate Editor. The reviewer's comments (not including confidential comments to the Editor) and the comments from the Associate Editor are included at the end of this email for your reference. As you will see, the reviewers and the Editors still raise some concerns with your manuscript and we would like to invite you to revise your manuscript to address them.

On the plus-side, the new version shows significant convergence; on the minus-side, doubts remain and new problems have been introduced in your revision. The latter concerns, for example, the use of 'synchrony'. I appreciate that you have made this change in response to earlier comments, but it is an unfortunate change. I feel you cannot adapt a word that has a clear interpretation in the mind of the reader and redefine it for your use. Clearly, synchrony is more than having the same periodicity. Using that word is therefore confusing and suggestive of more than can be justified by the results. Several other issues need to be mentioned clearly in a discussion, for example, remark 2 by the Associate Editor and the remark on the CFR-estimates by the reviewer. Although you should discuss remark 2 on the interpretation of Figure 3B, I also agree with the Associate Editor that it seems strange to call 3B a bifurcation diagram. There is no system state of which the stability-regime is assessed as a function of a changing system ingredient. Please clarify or change the way you refer to the plot. All other remarks should be addressed, but mainly asks for some additional clarification and some minor rewriting.

As you have realized by now, we have made an exception in your case to allow another round of revision (reason: we like the work, but we have not yet reached the level where everyone is unconditionally convinced of its findings). However, if profound doubts remain this could still lead to rejection. So I urge you to make every effort to fully address all of the comments at this stage. If deemed necessary by the Associate Editor, your manuscript will be sent back to one or more of the original reviewers for assessment. If the original reviewers are not available we may invite new reviewers. Please note that we cannot guarantee eventual acceptance of your manuscript at this stage.

Research ethics:

Use of animals and field studies:

All supplementary materials accompanying an accepted article will be treated as in their final form. They will be published alongside the paper on the journal website and posted on the online

figshare repository. Files on figshare will be made available approximately one week before the accompanying article so that the supplementary material can be attributed a unique DOI. Please try to submit all supplementary material as a single file.

Please submit a copy of your revised paper within three weeks. If we do not hear from you within this time your manuscript will be rejected. If you are unable to meet this deadline please let us know as soon as possible, as we may be able to grant a short extension.

Best wishes,
 Professor Hans Heesterbeek
 Editor, Proceedings B
 mailto: proceedingsb@royalsociety.org

Associate Editor
 Board Member: 1
 Comments to Author:
 Two key points from me:

1) The referee raises the use of the word "synchrony", and I strongly agree: to say two periodic things were in synchrony must surely mean both the same period and the same phase, i.e. they happen at the same time. When it's only known to be the same periodicity, it does not follow that they are synchronous. This should be resolved in the wording throughout (and not by redefining the word "synchrony" see some instances below).

2) The result that the dynamic regime depends on phase is still puzzling (figure 3B - which I don't think really is a bifurcation diagram). If it's just a seasonally forced SEIR model then the system should be invariant to phase - the whole thing should shift so if we're measuring rounded integer year of period, surely it should be invariant (fig 3b should be vertically constant). Is there something else hidden which is periodic (can't tell from supplement - are births a function of time), or could this just be an artefact of initial conditions or how periodicity is measured?

Some smaller points from me also, working in order through the current manuscript, mostly small typos or unclear wording resulting from previous edits:

line 75: "role of demographic on" -> demography surely

90: "based their" missing an "on"

116: "forwarded simulated" surely "forward simulated"

124: "regions locales", one or the other I guess

135: see above, this doesn't make sense as a definition of synchrony consistent with others' usage.

144-146: some awkward wording (edits not helped): "at given time" so coupling value is a function of time? "provides 1% of the infections in population X" isn't quite parsing right.

Looking at the model, it's σ of the infecteds act on the force of infection at the other place, so it's not 1% of infections, but maybe 1% of the new infections or infection rate.

154: "synchronous (i.e. same periodicity)"

191-194: not clear: "... in those regions each stochastic simulation... may not always biennial...". ?

Maybe "not all stochastic realisations were biennial"?

228: another i.e. redefinition

241: another redefinition of synchronised.
 247: synchrony through this paragraph

Reviewer(s)' Comments to Author:

Referee: 2

Comments to the Author(s)

The authors document measles time series of diverse dominant periodicity for London regions, estimate important model parameters from the data with the aid of demographic information, and reproduce to some extent the observed patterns by incorporating specific coupling rates and sinusoidal phase and amplitude transmission estimates into the model. The data set analysed is rather unique and their findings interesting to the field, but I strongly discourage the use of the term “synchrony” to refer to a coincidence in interepidemic period, simply because they are not interchangeable: biennial time series, for example, may present outbreaks in alternate years and thus be in opposite phases, a situation we would like to distinguish from biennial epidemics that peak simultaneously. Moreover, the authors later make reference to previous work that examined changes in synchrony among boroughs and cities as measured by the correlation between time series, further demonstrating that calling different notions the same leads to confusion.

Another aspect that could be discussed further is why is the importation rate assumed to be constant, when a fraction of the number of infectious individuals in the other patch at each time step could be used instead. Related to that, the text in [146] states “we used the seasonal transmission values” which is not the number of infectious individuals in the other population and I therefore do not find justified.

It does not make much sense to me that forcing phase plays the same role as initial conditions, could the authors provide more detail to support this argument?

Minor points:

[28] What is meant by “phase coupling”?

[89] The statement “the outer ring of London was not formally divided into regions” makes me wonder whether the mortality records used in this study were reported in the groupings used in this paper, or whether they were grouped in these regions by the authors. Which was the case?

[95] What resolution in time is assumed for the interpolated data? How are the interpolated data employed?

[114] The authors do clarify in their response to referees that they performed 60 simulations for each of the 400 sets of parameters used, but this not as clearly stated in the paper.

[123] Is the transmission rate “based on” or equal to the mean of their estimates?

[138] Could you provide more details as to how was regional isolation “relative” in the previous model? Were modelled populations not completely independent?

[182] How is population size related to interepidemic period (other than by defining whether the population is above the CCS)?

[196] I am unconvinced of the source used for CFR estimates. The cited paper (27) uses data from 1980 onward, while the paper under review analyses a data set from 1897-1906. Why would the CFR be similar in such different eras? It has definitely gone down since the turn of the 20th century, and in the 1980s plenty of treatments and medical knowledge were available that were not over eighty years earlier.

[198] Please discuss how incomplete reporting would modify these estimates.

[234] Please include references to the birth rates of each region in your discussion.

[277] What is meant by a “bifurcation grid”?

[304] I would expect lower school attendance to lower forcing amplitude, rather than modifying its phase.

Author's Response to Decision Letter for (RSPB-2019-1510.R1)

See Appendix C.

RSPB-2019-1510.R2 (Revision)

Review form: Reviewer 4 (Ryosuke Omori)

Recommendation

Accept with minor revision (please list in comments)

Scientific importance: Is the manuscript an original and important contribution to its field?

Good

General interest: Is the paper of sufficient general interest?

Good

Quality of the paper: Is the overall quality of the paper suitable?

Good

Is the length of the paper justified?

Yes

Should the paper be seen by a specialist statistical reviewer?

Yes

Do you have any concerns about statistical analyses in this paper? If so, please specify them explicitly in your report.

No

It is a condition of publication that authors make their supporting data, code and materials available - either as supplementary material or hosted in an external repository. Please rate, if applicable, the supporting data on the following criteria.

Is it accessible?

Yes

Is it clear?

Yes

Is it adequate?

Yes

Do you have any ethical concerns with this paper?

No

Comments to the Author

This manuscript is quite interesting in terms of both epidemiology and complex system. Previous studies have been pointed out that the possibility of coexistence of multiple attractors in the system which the force of infection is seasonally varied, this paper suggests example observed in real world. I really enjoyed reading except one question: the authors showed legion specific periodicity because of the estimate of coupling rate is small. I am a bit surprised that biannual oscillation was observed for a long time even though immigration events (equivalent with coupling) should have happened. If considerable explanation can be existed, please discuss.

Decision letter (RSPB-2019-1510.R2)

20-Mar-2020

Dear Mr Becker

I am pleased to inform you that your manuscript RSPB-2019-1510.R2 entitled "Coexisting attractors in the context of cross-scale population dynamics: measles in London as a case study" has been accepted for publication in Proceedings B.

The referee has recommended publication, but also suggests some minor revision to the discussion in your manuscript. Therefore, I invite you to respond to the referee's comments and revise your manuscript. Because the schedule for publication is very tight, it is a condition of publication that you submit the revised version of your manuscript within 7 days. If you do not think you will be able to meet this date please let us know.

- 1) A text file of the manuscript (doc, txt, rtf or tex), including the references, tables (including captions) and figure captions. Please remove any tracked changes from the text before submission. PDF files are not an accepted format for the "Main Document".
- 2) A separate electronic file of each figure (tiff, EPS or print-quality PDF preferred). The format should be produced directly from original creation package, or original software format. PowerPoint files are not accepted.

3) Electronic supplementary material: this should be contained in a separate file and where possible, all ESM should be combined into a single file. All supplementary materials accompanying an accepted article will be treated as in their final form. They will be published alongside the paper on the journal website and posted on the online figshare repository. Files on figshare will be made available approximately one week before the accompanying article so that the supplementary material can be attributed a unique DOI.

Sincerely,
 Professor Hans Heesterbeek
 Editor, Proceedings B
<mailto:proceedingsb@royalsociety.org>

Reviewer(s)' Comments to Author:

Referee: 4

Comments to the Author(s)

This manuscript is quite interesting in terms of both epidemiology and complex system. Previous studies have been pointed out that the possibility of coexistence of multiple attractors in the system which the force of infection is seasonally varied, this paper suggests example observed in real world. I really enjoyed reading except one question: the authors showed legion specific periodicity because of the estimate of coupling rate is small. I am a bit surprised that biannual oscillation was observed for a long time even though immigration events (equivalent with coupling) should have happened. If considerable explanation can be existed, please discuss.

Author's Response to Decision Letter for (RSPB-2019-1510.R2)

See Appendix D.

Decision letter (RSPB-2019-1510.R3)

30-Mar-2020

Dear Mr Becker

I am pleased to inform you that your manuscript entitled "Coexisting attractors in the context of cross-scale population dynamics: measles in London as a case study" has been accepted for publication in Proceedings B.

Open Access

Paper charges

Sincerely,
Editor, Proceedings B
mailto: proceedingsb@royalsociety.org

Appendix A

Response to Referees for Manuscript RSPB-2019-0213 entitled "Coexisting attractors in the context of cross-scale population dynamics: measles in London as a case study"

Dear Dr. Heesterbeek, the Associate Editor, and Reviewers,

We are pleased to submit our revised paper "Coexisting attractors in the context of cross-scale population dynamics: measles in London as a case study", for consideration to be published in *Proceedings of the Royal Society B*.

We would like to thank you and the reviewers for their careful, considered, and very helpful comments. We are additionally thankful to have a chance to resubmit our manuscript. In the letter that follows, we will respond to comments (shown in...) and outline our changes (shown in...).

We appreciate your consideration of our work for publication and the chance to resubmit our manuscript. Please let us know if you have any additional questions or concerns and we look forward to hearing from you.

With best regards,

Alex Becker, Susan Zhou, Amy Wesolowski, Bryan Grenfell

Editor:

*In addition to the comments by the reviewers and the Associate Editor, I want to make a remark on John Brownlee. You write in lines 71-72 that John Brownlee "lacked the mechanistic framework now available to researchers". This is not entirely correct. Of course the field has advanced significantly, but the main reason why Brownlee could not do such analyses was a fundamental difference in how he approached the subject and how he felt epidemic phenomena should be explained. See a paper by Paul Fine (1979 'John Brownlee and the measurement of infectiousness ...', *J. Roy Stat Soc A*, 142, 347-362) and my own paper 'The law of mass action in epidemiology: a historical perspective' in Cuddington & Beisner (Eds.) 'Ecological paradigms lost: routs of theory change', 2005. Brownlee approached the topic a posteriori (by fitting curves to outbreaks that had occurred) and thought that reduction and gain of infective power by the infectious agent were the mechanism behind outbreaks. This in contrast to notably Ronald Ross, who worked a priori from basic assumptions, and thought that it was the availability of susceptibles that should be the mechanistic basis. Ross brought forward his approach clearly in 1911 and again in a series of publications from 1916. He and Brownlee had a polemic about the approaches and ultimately, Brownlee conceded that his approach and view of the matter was not the correct one. I'm not suggesting that you go into all this (or even refer to the above papers), but perhaps the wording can be slightly adapted. Brownlee certainly has a theory and considerations, they were just not the theory and considerations that survived in history to what we have today. That alle stemmed from Ross' approach developed concurrently with Brownlee's approach (and well-known to him). One can only imagine what insights Brownlee could have gained with his data had he adapted Ross' approach from the beginning.*

Thank you for this note, it's very interesting to see how the approaches evolved. We've changed the wording to indicate that Brownlee's approach differed from the mechanistic framework we often use today. Hopefully this is a more accurate portrayal.

Associate Editor:

This is very nice work, presenting a dataset which is unusual in breaking down measles dynamics within city pre-vaccination. There's a story with differing dynamic regimes between regions of London (1- or 2- year periodic), yet the aggregate would appear as simply annual dynamics - this is surprising given that one would expect the dynamics to be so closely coupled, but the authors present a plausible explanation of this in a modelling framework.

Both referees raise points to address. Referee 1 raises some major points, particularly focusing on clarification of what was done. Referee 2 gives an extensive list of minor points for clarification also.

We would like to thank the Associate Editor for a positive initial evaluation, and are grateful for the following comments that we have addressed.

- surprised not to see comment on using mortality data: does this add any issues, e.g. seasonal bias over case data as CFR changes?

We have emphasized this in the discussion by clarifying a point (with reference to Mantilla-Beniers 2010), and by adding the following:

Further, although the monthly region-level incidence data analyzed in this manuscript generally exhibit clear periodicities, mortality data may lag slightly behind the true incidence counts (32), leading to potential biases in estimates of seasonal phase. However, these biases are likely to be consistent across locations. Additionally, by assuming a constant CFR we are effectively treating the mortality data as scaled incidence data.

We have also clarified in the Methods section that the CFR is time-invariant.

- the SI format of the word document mangles some of the equations. Maybe safer to submit it as PDF.

We apologize for any unclear equations and have now included a PDF version.

- as one of the referees mentions, could do with being more upfront about data limitations (e.g. in fig 1, outer South looks super noisy, and outer North is not at all convincingly annual)

Thank you for your comment. We have added this to the results section noting that some regions, particularly the Outer Ring, were noisy, possibly due to lower overall mortality counts and therefore limiting our analysis of these regions.

- Fig S8 (two patch sims), there's background grid lines for years, but make the axes ticks also integer number of years

This has now been changed in S8.

- Penultimate paragraph of main manuscript, can the differing birth rates (as shown in fig 1 region colours) be discounted as being the key difference between the regions driving the dynamics?

Birth rate is certainly important to the dynamics; however, we argue that variation in seasonal transmission has a larger impact. We have expanded on this in later Reviewer 1 comment, but we have also added a Principal Component Analysis to look at the how much variation is explained by birth rate, seasonal amplitude, and seasonal phase. This has also been added to supplement. We find that approximately 70% of the variation can be explained by PC1 of which amplitude and phase play the largest role (0.64 and -0.66, respectively). Although birth rate is not unimportant (0.4) on PC1, the seasonal patterns yield a larger effect. Indeed, birth rate is more important in PC1 (0.91 versus -0.35 (amplitude) and 0.22 (phase)), however this axis in contrast only explains about 27% of the variation.

From this additional analysis, we can conclude that between birth rate, amplitude, and phase, the seasonal patterns are the larger driver of periodicity. This of course does not mean birth rate is unimportant, however if we are qualitatively looking for a simple explanation, seasonality provides a more parsimonious one.

Referee 1

This is a very interesting paper, and potentially has lots of powerful insights - but these are unfortunately hidden by a lack of detail. I appreciate that there are word limits to the publication, but much could be communicated to the reader without adding too much extra text.

We thank the Reviewer for their helpful comments, which we address below. We've broken down the major comments into subparts here for clarity.

1a) The model is fitted using the methods of (28,29). This tells the reader very little. Is this fitting the deterministic or stochastic version of the model.

We have clarified that we fit the stochastic SEIR model. We have added a brief summary of the iterated filtering process in the methods section. Notably, we outline the primary function of the algorithm, e.g. allowing parameters of interest to take random walks in order to maximize likelihood. We have added significantly to the supplement, in particular about the model, likelihood function, and fitting method.

1b) What statistically characteristers a "good fit".

We would like to thank the review for this comment (which was also raised by Reviewer 2).

Statistically, a “good fit” is the parameter set that yields the maximum likelihood after an exhaustive sweep of parameter space (e.g. using iterated filtering) under the assumption that the chosen model (e.g. the SEIR model with measurement and process error) is the correct one.

Although methods such as least squares provide a number for the error between the observed and simulated model, this metric is questionable for stochastic systems such as epidemic dynamics. This issue is further compounded by using the possibly more stochastic death data, due to the lower values observed in comparison to incidence data.

Perhaps the most straightforward, and likely well-documented, method in the literature is visual fit. This yields insight into the general dynamics such as epidemic shape, timing, and size. However, this is challenging to quantify and is dependent on each individual’s views (e.g. should the visual fit generally capture trends, or should it go through the majority of data points). While we cannot fully solve those problems by showing 95% quantiles (the blue ribbons shown in Figure 2), we can say with confidence that in, for example the West region, the data is well within 95% of the fitted model’s behavior. This is a significant improvement over showing deterministic means.

Further, we choose to quantify the fit by comparing the simulated fitted model periodicity against the observed data. While this again has some level of visual component, we can compare the peaks of the power spectra. For example, the mean of the stochastic simulations in the Central region for period one makes up approximately 15% of the power spectra. This agrees well with the data, at again about 15%.

Lastly, we looked at percent of out of phase simulations for the pairwise analysis. Here, we can say 85% of the North-West pair of simulations are out of phase and check this against the data North-West are out of phase for the identified parameters.

We feel that more exhaustive, and perhaps “standardized” methods of accessing fit, are needed and is an area of future research in the field, although beyond the scope of this manuscript.

1c) I was surprised that imports of infection are not mentioned - what stops the stochastic model from going extinct?

In addition to the above comments, we also clarified that we do estimate a yearly importation rate in the Methods section. We additionally have added a table of parameter estimates to the Supplement (and shown below in this letter).

2) Similarly, it is stated that "We then estimated the coupling value for ...". How? This is a very difficult problem, especially if the coupling is relatively low, or the populations are closely synchronised. Much more detail is needed.

We have now clarified the iterated filtering algorithm that we used to profile over the range of mixing rates while using the region-specific values obtained in the local region analysis such as seasonality and initial conditions. We have now provided a more exhaustive summary of the iterated filtering method in the Supplement.

3a) The different regions have different amplitudes and phases (described as "higher phases" on page 4 - a term I don't think informs).

We have changed "higher" to "greater".

3b) While the difference in amplitude has some mechanistic basis (poorer regions may be more impacted by seasonal weather), there is no mechanistic argument for the very different phases.

We speculate in the discussion that variation in the school calendar likely contributes to these differences in phase, however, we were unable to find consistent historical documentation among London regions supporting this. Generally, historical school calendars are challenging to find even at very small subscales such as sub-boroughs. We have clarified in the discussion that while we could not find such documentation, each seasonality approximately following the school (peaking in Feb / March and again in Sept / Oct, with declines in the summer months) supports a school calendar mechanism.

3c) In fact, I'm confused if the authors are arguing for birth rate leading to the dominant behaviour, or whether it is the seasonal forcing. These two factors are never disentangled.

Thank you for this point, as it is important and one that the Associate Editor raised as well. We argue that seasonal parameters, particularly phase and amplitude, lead the behavior as opposed to birth rate. We explored this using a Principal Component Analysis (as discussed above in a reply to the AE). Notably, we found the PC1 explains 70% of the variation with both phase and amplitude playing a larger role than birth rate. We have updated the main text and supplement to include this analysis.

3d) Moreover, I cannot see any reason why the phase of the forcing would have any impact on the dominant periodicity (figure 3). Surely the phase simply shifts the outbreaks in time, but don't change their period?

We apologize for any unclear language in the text. We have now attempted to clarify this in the discussion section. Overall, phase acts in a similar way to initial conditions.

One way to think about it is in terms of the initial effective transmission, $R_e = \frac{\beta S(0)}{N}$.

Earn et al (Science, 2000) showed that variation in initial conditions, particularly variation in the starting susceptible dynamics can lead to different basins of attraction in the moderation transmission region. Here, while we find very similar initial conditions (table of all parameter estimates are shown in the Supplement and in a Reviewer 2 comment), variation in phase acts in a similar way. However, since phase annually repeats, we are able to disentangle these two parameters.

Notably, we can say

$$R_e(t) = \frac{\beta_t S(t)}{N} = \bar{\beta} \frac{(1 + \alpha \sin(2\pi t + \phi))}{N} S_t$$

Where then at $t = 0$

$$R_e(0) = \frac{\beta_0 S(0)}{N} = \bar{\beta} \frac{(1 + \alpha \sin(2\pi \cdot 0 + \phi))}{N} S_0 = \bar{\beta} \alpha \sin \phi \frac{S_0}{N}$$

Assuming $S(0)$ is similar between regions (range: 0.04 – 0.066), the impact of varying phase between approximately 0.5 (annual) and 1.5 (biennial) will result in a larger impact than $S(0)$. Furthermore, and importantly so, as phase will impact the dynamics each year, it continues to modulate the effective transmission rate past $t = 0$.

4) Looking at the power spectra in the Supplement, it is clear that some regions show relatively poor agreement between models and data (inner South, outer West). It would be good to be more honest about this lack of fit in the main paper, and speculate as to the reason.

We have now expanded on this in the Results section.

1) Crude Birth Rate needs to be defined.

We have added a definition to the Results section, as well as on expanded on what the ranges of CBR are in the data set.

2) Page 4 "two shape transmission parameters" should be "two dimensional transmission parameters"

Thank you, the text has been updated.

2) Bottom of page 4. I don't understand the phrase "population mixing in between region synchrony"

Reviewer 2 also commented on this, we have removed the word 'between' and tried to clarify the sentence by stating:

Thus far we've analyzed local dynamics to gain insight into the drivers of measles periodicity; we now turn to the potential role of population mixing on epidemic synchrony

3) Lots of the references have { } in the middle !

Apologies for the formatting issue, we have now corrected these errors.

4) *Figure 2, why is 1898 a larger epidemic year than all the others - some biological explanation is needed.*

It is difficult to know why 1898 was a larger epidemic year without having the mortality or birth data prior to 1897 (we were unable to find registrar general reports prior to this year). We searched the London boroughs medical records to find references. We found two of note, primarily indicating that this was unusual even in the context of prior to 1898.

In 1898, the medical officer of Islington wrote “measles was unusually prevalent and fatal...in one year only since 1885 were so many deaths registered from measles, namely in 1887 when 335 were entered.” (1)

The medical officer of Wandsworth concluded similarly, stating “a very violent epidemic of measles raged during the months of April and May [1898]”. (2)

We can speculate that there was a larger than unusual birth cohort up to four years prior that led to a uniformly larger than expected outbreak, but without the data or more detailed medical records, it is very difficult to determine the likely mechanism.

(1) <https://wellcomelibrary.org/moh/report/b17998785/53#?m=0&cv=53&c=0&s=0&z=-2.0486%2C-0.1192%2C4.6355%2C1.8096>

(2) <https://wellcomelibrary.org/moh/report/b18250671/179#?m=0&cv=179&c=0&s=0>

5) *Figure 4 captions say "Note the different x-axis on the top row" - I think it would be clearer to say "The top and bottom rows have different scales on the x axis".*

This has been corrected.

6) *The figures within the supplement do not tie-up with the text. For example the text refers to Figure S5A and S5B, whereas the figures are actually S8A and S8B*

Thank you and corrected.

Referee 2:

Becker et al investigate the standard assumption of homogeneous mixing between city regions aided by a detailed dataset of measles mortality in London. This dataset had been studied nearly a century ago by John Brownlee, in a paper cited by the authors. As they remark, Brownlee did not have the modelling framework —and the computational power, I would add— that is exploited by Becker et al to better explain differences in periodicity observed between regions of London in spite of presumably high coupling between regions.

After characterising the periodicity of measles mortality time series in five inner and four outer regions of London, the authors estimate yearly mean regional transmission rates, together with

their amplitude and phase. In turn, parameter estimates are used in modelling, and the periodicity found in fitted models and disease mortality is contrasted.

Later on, Becker et al use their estimates of yearly transmission parameters to estimate coupling rates in a two-patch metapopulation model fitted to two pairs of time series. This allows them to study the percentage of model time series that are out of phase for each coupling parameter. They find that the phase and amplitude that characterise forcing in each region may define whether time series are out of phase even at high coupling rates.

The issues analysed in the paper under consideration are of key importance in our understanding of disease spread and are difficult to investigate due to the scarcity of data of sufficient spatial and temporal resolution. Moreover, the studies performed in this paper are generally thorough and convincing. I strongly recommend its publication after some minor issues are solved.

We thank the reviewer for positive comments as well as a number of specific points which we believe have improved the manuscript.

Specific points and typos

We thank the reviewer for the following specific points and we apologize for any inconsistencies in word choice as well as any typos.

1. Both “test bed” and “testbed” are used in the text. Please use one only.

Thank you, we have corrected to just be “testbed”.

2. I suggest using only the term coupling parameter; “mixing” is occasionally used in the text as a synonym, but it is confusing.

We have changed the vast majority to strictly coupling. Note Figures 4 and S6-9 have been adjusted to account for this.

3. Include a table of parameter estimates for all regions.

This is an excellent suggestion, and one that ties into an earlier comment about importation rate. I have included the parameters estimated in the model (note that b_i is related to transmission R_0). We have included this into the supplement, as well as below.

	R_0	b_1	b_2	b_3	b_4	b_5	b_6	σ_{SE}	ι	ψ	S_0	E_0	I_0	ϕ
All	20	7.05	7.707	7.333	7.184	6.95	7.59	0.035	367.5	0.153	0.054	7e-05	6e-05	0.016
South	18	6.945	7.485	7.137	6.945	7.166	7.45	0.036	56.56	0.235	0.058	7e-05	6e-05	0.016
West	18	7.177	7.304	7.311	6.751	7.277	7.219	0.018	4.71	0.312	0.066	5e-05	5e-05	0.016
Central	30	7.496	7.878	7.915	7.729	7.105	7.959	0.066	15.41	0.276	0.045	5e-05	5e-05	0.017
East	31	7.424	7.961	7.833	7.76	7.273	8.007	0.041	58.23	0.249	0.040	4e-05	4e-05	0.017
North	20	7.309	7.43	7.457	7.089	6.983	7.389	0.04	52.96	0.215	0.059	5e-05	5e-05	0.015
Outer west	16	6.646	7.501	7.013	6.782	7.091	7.339	0.069	6.335	0.389	0.065	6e-05	5e-05	0.009
Outer east	25	7.241	7.905	7.578	7.752	6.71	7.923	0.098	80.43	0.177	0.042	4e-05	5e-05	0.013
Outer south	23	7.325	7.527	7.57	6.993	7.664	7.45	0.076	6.48	0.2	0.048	5e-05	5e-05	0.009

Outer north	27	7.393	8.146	7.401	7.539	7.357	7.693	0.098	22.62	0.237	0.047	6e-05	5e-05	0.01
-------------	----	-------	-------	-------	-------	-------	-------	-------	-------	-------	-------	-------	-------	------

4. Line 22 Says “underling” instead of “underlying”.

We have corrected the text to say “underlying”.

5. Line 47 What is meant by “density” of human mixing?

We have clarified here that we mean the aggregation of human populations.

6. Line 55 Does “synchrony across local population” refer to insignificant phase differences between city-level subdivisions?

Yes, we have aimed to clarify this.

7. Line 56-57 Is the period truly approximated by the dominant period of the largest subpopulation, or by the period of the whole population?

Thank you, it should be the overall population. This sentence has been clarified.

8. Line 110 In what sense was the deterministic model simulated? Was it integrated numerically, for each parameter set? Please clarify and provide the details.

We have clarified that the model was integrated numerically.

9. Line 111 If this is the analysis shown in figure 3, please refer to figure. If results are not shown, please state this.

Thank you, we now reference this figure in the Results section.

10. Line 130 Please detail how coupling parameters were estimated and the way in which phase differences were quantified.

We have clarified here that coupling parameters were estimated using profile maximum likelihood. Additionally, we have added more detail about the fitting process and model to the Supplement.

11. Line 137 Starting year should be 1897. Maybe write “from 1897 to 1906”, instead of using the dash?

Apologies for the typo here -- we have changed the text to say “From 1897 to 1906”.

12. Line 159 How did the model capture historical CFR? Please explain whether and how it was estimated.

We have clarified this sentence to include that this is from a meta-analysis of CFRs at the nationally-representative scale.

13. Line 169 *Similar to my comment to line 110, in what sense was the deterministic model simulated?*

We have clarified that we numerically integrate the deterministic model.

14. Line 174 *Should read: “one-, two- and greater than two-year”.*

Thank you, this has been corrected.

15. Line 179 *Should read: “predictive”.*

This has been corrected.

16. Line 181 *Should read: “to gain insight”.*

This has been added, and the sentence clarified as per the next comment and Reviewer 1’s.

17. Line 182 *Maybe omit the word “between” from the sentence?*

Thank you, we have removed this, and also clarified the sentence per Reviewer 1’s comments.

18. Line 242 *Should read: “explains”.*

This has been corrected.

19. Line 181 *Omit comma after “analyzing”.*

This has been omitted.

20. *Figure 1 How well connected internally is each region? The South seems much larger, and probably less dense than the other inner regions. The Outer East is cut in two by the river, it seems odd to consider it an epidemiological subpopulation.*

This is an excellent point, however difficult to answer. Early maps of the London Underground were not available until 1907 (as far as we are aware), making comparison with this 1897-1906 data challenging, and look like this map from 1908:

Obtained from: <https://londonist.com/2016/05/the-history-of-the-tube-map>

These maps do not typically contain information on the Outer Ring, and it does not appear that the Underground was extended to these regions until later eras. In similar fashion, although the London Poverty maps generated by Charles Booth (<https://booth.lse.ac.uk/map/12/-0.1583/51.5098/100/0>) are informative, they do not extend to the traditional Outer regions. We were also unable to find movement data or historical street between regions that could be used as a proxy. However, there was a boom in the number of bridges and roads built in the mid 19th century, indicating that although a region like Outer East is divided by the river, it may still be reasonably well-mixed.

A future direction with the London Borough data is to attempt to infer connectivity based on transmission and currently unpublished methods from this team and colleagues.

21. Figure 3A The legend is unclear: I cannot tell West from Outer South or Central from Outer North. Gray and pale blue are also very hard to distinguish from black and blue, and I cannot find the dot-dashed key on the legend.

We have reworked Figure 3 for clarity. In the top panel, we have removed variable line types, and instead have the raw seasonality points shown for each region color coded by periodicity. We use this periodicity as a grouping (either annual or biennial) and show the mean seasonality \pm SE for each group. We feel this averaging still illustrates the substantial difference in phase and amplitude between the two groupings without the additional line types or color coding by region type.

In line with this, we have therefore removed the labels (A-F) from Figure 3B. We have also changed the color scheme in Figure 3 to match that in Figure 1 (it was previously flipped).

Supplementary material

1. Draw a line in figures S1, S2, S4, S5 to indicate which values correspond to statistically significant power.

This is unfortunately a challenge since in order to create a confidence interval we must fit a model to the power spectrum. This is okay for “nice” data, but begins to create problems with “noisy” data. We have attached two plots that will hopefully explain this difficulty. In both figures, the solid black line refers to the power spectra calculated directly from the data, the dashed black line refers to the fitted power spectra, and the red line indicates 95% confidence intervals in which values above this line are significant. For example, for the Aggregate time series below, a periodicity of 1 year is significant.

In the Inner London data (shown below), the majority of true power spectra and model power spectra agree, although there are some key differences. For example, in the West region, the power spectra model under predicts the annual proportion and in the model, this is not significant, but in the data, it would be. Note that when we say “model power spectra” in this answer, we are referring to the fitted power spectra, *not* our fitted and simulated SEIR model.

In the Outer London data (shown below), these errors are more pronounced between the fitted power spectra and the true power spectra. While the model power spectra and true spectra mostly agree in Outer West (OR.west), the other regions experience significant departures from the data. In these three outer regions, it is difficult to compare the fitted power spectra against the data, drastically limiting our ability to draw any conclusions about significant here.

Due to this, we are hesitant to compare the confidence intervals drawn from a spectra model to both the data and our stochastic simulations from the SEIR model.

2. Revise the expression “albeit slightly noisier”. Maybe rephrase as “even if slightly noisier” or “albeit in a noisier fashion”?

This has been rephrased to ‘even if slightly noisier’.

3. The fit of model time series to data, or to data behavior (periodicity) is not quantified. Confidence intervals for one and the other are shown as a visual test of the goodness of the fit.

We address a similar comment above from Reviewer 1 (please see above as well). We believe that standardizing this across the field would be a great addition and point of clarification. However, we feel that this is beyond the scope of the current manuscript.

One advantage of visual measures such as the 95% quantiles shown in the resimulated model (Fig 2) and the power spectra fits (S4) is that it allows individuals to assess goodness of fit based on their own measures. For instance, we can say for the aggregate data, about 20% of the power spectra goes to biennial patterns, which the simulations agree with well (although there is variation in the individual simulation behavior), thereby giving some

metric of periodicity between data and simulation. However, we believe this is more informative than summary statistics as it allows the viewer to draw their own conclusions (e.g. if they want to assess agreement between annual proportions of the data and simulation for a biennial region).

4. Line 57 “parameters matched well with the predicted dynamics, which showed strongly in phase dynamics, well. ”

We have now corrected this typo.

5. Line 92 It may be possible the that stochasticity,...”

We have now corrected this error.

6. Figure S6 Please clarify whether the percentages shown are taken for each mixing rate, and not over all possible rates used.

Thank you, we have now clarified this in the caption.

Appendix B

Response to Referees for Manuscript RSPB-2019-1510 entitled "Coexisting attractors in the context of cross-scale population dynamics: measles in London as a case study"

We greatly appreciate the opportunity to respond to this second round of reviews for our manuscript, "Coexisting attractors in the context of cross-scale population dynamics: measles in London as a case study".

The manuscript has now received four reviews, and we have substantially incorporated this feedback as part of a revised analysis that models the seasonal transmission as a sinusoidal function as opposed to a more complicated spline function. We appreciate that *Royal Society Proceedings B* does not typically conduct multiple rounds of revision, so we have attempted to address each point thoroughly. However, if there are areas of additional clarification, we are happy to address these further.

We would like to thank you and the reviewers for their careful, considered, and very helpful comments. We are additionally thankful to have a chance to resubmit our manuscript. In the letter that follows, we will respond to comments and outline our changes.

We appreciate your consideration of our work for publication and the chance to resubmit our manuscript. Please let us know if you have any additional questions or concerns and we look forward to hearing from you.

With best regards,

Alex Becker, Susan Zhou, Amy Wesolowski, Bryan Grenfell

Response to referees:

Referee: 2

We thank the referee for their continued comments and suggestions.

We want to point out a major change upfront based on comments from yourself and Referee 3: we have changed the inference model to be a sinusoidal transmission function as opposed to a six-point spline. We believe this not only simplifies the description of the fitting process, but it indeed strengthens the results. We also want to point out that we have changed the y-axis of Figure 4 to be “percent synchronized” instead of “percent out of phase” (terms explained below, and in the MS).

2.1) Thank you for the details on the fitting procedure. I would appreciate if the authors could distinguish exactly what parameters are fitted at each stage: are monthly transmission rates estimated alongside CFR, importation rate and “measurement” error, with the stochastic SEIR? How is the sinusoidal transmission function fitted?

We have clarified (below) in the methods section that all parameters are fit in tandem for the iterated filtering portion.

“Briefly, the iterated filtering method aims to maximize likelihood by allowing all parameters of interest to take random walks in tandem.”

In our new formulation, the sinusoidal transmission function is fit directly to the data at the same time as the other parameters.

2.2) I presume monthly transmission rates are fitted with the stochastic SEIR (though it seems like a lot of parameters are fitted at once), and those estimates are then fitted using cubic splines. What exactly are the coefficients b_i given in the table? I am also unsure of how the amplitude and phase of the sinusoidal transmission function are estimated from the monthly rates; monthly estimates appear to have more than one maximum for the biennial regime, which would suggest the sinusoidal function may need a modification in frequency too -unless the authors are picking the phase and amplitude that lead to epidemic cycles of the observed periodicity.

We have clarified that seasonality is now modelled as a sine transmission function and as part of the iterated filtering algorithm, all parameter are fit in tandem.

2.3) The authors explain that the deterministic SEIR is integrated numerically in order to obtain figure 3; I recommend referring to this figure in that part of the text.

We have now corrected the reference in the main text.

2.4) I thank the authors for including a table of parameter and initial condition estimates. Is it possible to include an explanatory text for the table? I would suggest leaving phase and

amplitude estimates to another table, or marking in some way that they are the result of another round of fitting.

We have added some additional text to this (below). Additionally, as now seasonality is modeled as a sine variable and the language has been corrected.

“In the table that follows, we show the estimated parameters for each location included in our analysis. Parameters refer to the above SEIR model description. Our inferred parameters broadly fell in line with previously analyses. We found a mean transmission rate, R_0 , of 22, similar to the range described by (1, 4, 5). Initial conditions and multiplicative noise were similar to a later-era measles analysis (1950-1966) using the same framework (1). However, we did find a higher mean importation rate than previous analyses (again in the 1944-1966 era). Notably, we found a strong population dependence on the inferred importation rate ($R^2 = 0.9$, p-value $< 1e-4$), suggesting this may be a genuine feature of the data in this era.”

2.5) The phase of two different things may be relevant to this work, but they are not always clearly distinguished: one is the phase of the sinusoidal forcing function; the other is the epidemic phase of a given time series, and the paper does not explain whether it is quantified and how. Moreover, I am not convinced of using epidemic phase to discuss epidemic synchrony between populations: I understood that the authors seek to demonstrate that different dynamic regimes (e.g., annual and biennial epidemic cycles) can coexist in the presence of intermediate or high coupling rates, not whether outbreaks are in phase. If epidemic phase is indeed what they want to compare, please explain the methodology used to do so.

We have clarified that we are seeking, as you mention, to demonstrate that different dynamic regimes can coexist. We have changed the wording from “phase” to “synchrony” or “synchronized” we further clarify our definition of these terms in the Methods section, as below:

“To investigate ... region synchrony (here, this is defined as two regions with the same periodicity)”

We believe this is clearer and more consistent, and reserve “phase” for just the results of the seasonality discussions.

2.6) I would prefer to see the confidence levels shown in the response in the paper; to me, showing them does not subtract from the results. Along those lines, I suggest commenting on the coupling rate estimates: according to figure S7, only two or three pairs have a coupling rate significantly different from 0.

Although the confidence intervals match generally-well to the observed regional power spectra, they differ greatly in the outer region, and therefore greatly limit the ability to draw conclusions from there. Additionally, as adding the confidence intervals requires another set of modeling fitting, we feel that adding this to the Supplement will add additional complexity and possible confusion to the manuscript.

We have clarified in the supplement that indeed only four regions have coupling rates significantly different than 0.

2.7) Four parameters receive duplicated labels (and I wasted some time realising this!): ϕ (CFR and phase) and σ (incubation and coupling rate). Could you please give them different labels?

Apologies -- fixed.

2.8) I was a little surprised to see such low estimates of case fatality rate for the era. Presumably, measles case fatality has declined over time.

As mentioned in the text, these values are in line with other historical estimates.

2.9) The lower graphs in figure 4 and S6 give the percent of simulations that are out of phase for various coupling rates. Please state that this percentage is taken over all simulations with that coupling rate (this may be obvious in the figure of the main text, but is not on the one that is in the supplement).

We have adjusted the caption of Figure S6 this to read:

“The percent of simulations which resulted in synchronized (i.e. same periodicities) dynamics for each coupling rate between East and Central London.”

2.10) Main text line number:

74] Maybe rephrase along the lines of: “utilized (24, 25) and necessary to estimate seasonal transmission parameters and test the role of demographic changes on epidemic dynamics.”

We have corrected the text to:

“Although Dr. Brownlee analyzed the cross-scale London data using periodograms (a method still crucial in the field (23)), his approach, based on a pathogen’s time-varying infectivity, differed from the mechanistic framework now commonly utilized in the modern era (24, 25) and necessary to estimate seasonal transmission parameters and test the role of demographic on epidemic dynamic.”

2.11) 86] and an approximate four outer

Fixed.

2.12) 109] Should read “forward simulated”?

Corrected.

2.13) 110] Refer also to the supplement for additional details on the fitting process.

We have added the following text here:

“A full discussion of the algorithm, and numerical implementation, can be found in (29, 30) with more details in the Supplement.”

2.14) 115] How are the values for phase and amplitude selected? It seemed like the transmission term of the fitted model was not sinusoidal.

These values are now part of the fitted model explicitly.

2.15) 189] “these two dimensional transmission parameters”

Corrected.

2.16) Supplement

Model description is now slightly mangled: it is mentioned that a stochastic version of the equations is used, but the equations include stochastic terms.

We have clarified that this model is fully deterministic when the noise term W is fixed with standard deviation = 0.

2.17) Please give more details of the fitting procedure that yields cubic splines for the seasonal transmission rate and, if adequate, list the coefficients for the cubic splines (x_i) that appear in its definition. Are the authors first finding estimates for the monthly transmission rate, and fitting these set of estimates with cubic splines?

As we have now modelled seasonality using a sine transmission function this should be clarified.

2.18) How well do model dynamics correlate when changing from the estimated monthly transmission rates to sinusoidal forcing?

As shown in the updated results, the dynamics stay largely the same. Measure of amplitude and phase remain very similar in this framework, and transmission R_0 has a very similar mean (albeit with a tighter range). The visual fits are very similar as well.

2.19) It is unclear which parameter values come from which fitting process, since both the coefficients b_i and the phase are given in the table, but I had understood they came from different stages of the fitting.

We have clarified that all parameters are now fit at the same time.

2.20) Estimates for R_0 seem extremely high, particularly for the Central and East regions, how do you interpret this?

Generally, these estimates, although slightly above the range given by Anderson and May (1991) via serological data (12-18), they are in line, and indeed, slightly, to greatly, lower than other transmission values obtained using time series data (ranging from 21-57 in He et al 2010). We have added a note on this in the Supplement text before the parameter table.

2.21) 6] Model description is now slightly mangled: it is mentioned that a stochastic version of the equations is used, but the equations include stochastic terms.

Addressed as above.

2.22) 22] I am confused by the equation used to fit βt , which I took to be the transmission rate in biweek t , and cannot see why a single number is fitted using cubic splines. Presumably, the splines really fit the annual variation in transmission (is this the case?), but then I would have expected to see some explicit dependence on time of the expression defining βt I take it the x_i depend on t . Is this correct? How does this connect to the sinusoidal transmission term mentioned in the main text?

As we are now fitting a sine transmission function there are no longer cubic splines involved.

2.23) 30] State upfront that underreporting could be estimated as part of the procedure, but it will be assumed that reporting is perfect. The phrasing is confusing at present. Also, since the expression for the parameters of the cumulative normal distribution is simplified by this assumption, it may be wise to write it in the equations of lines 33 and 38.

Fixed as below in the Supplement:

Although under-reporting could be estimated in this framework, we choose to fix it at 100% here as we assume mortality is well-reported.

$$P[\text{deaths} | D] = \Phi(\text{deaths} + 0.5; D, \psi^2 D^2) - \Phi(\text{deaths} - 0.5; D, \psi^2 D^2)$$

2.24) 33] Any reason why the value 0.5 is preferred?

We chose 0.5 to remain consistent with the He et al, 2010 analysis.

2.25) 41] Are parameters allowed to change simultaneously? (What is the dimension of the space in which these random walks take place?)

We have clarified that parameters are allowed to change simultaneously. The dimension of the space is therefore the number of parameters to be estimated.

2.26) 44] How do 60 iterations render 400 parameters? (This is probably not what the authors are saying, could they please clarify?)

We have clarified this to show we are iterated 400 different sets 60 times.

2.27) 45] Am I correct in thinking that you have a set of parameter sets?

At the end of inference this is correct, however we just choose the MLE.

2.28) 54] “due to aggregation of”... Disconnected populations?

Corrected to refer to population size.

2.29) 120] for each simulation set of parameters... to produce a set of theoretical predictions.

We have updated this sentence to read as follows:

“Using a range of amplitude (range: 0.03-0.35) and phase values (range: 0-2, corresponding to a maximum lag of approximately four months), we numerically integrated a deterministic SEIR model (equations show in the Supplement) for each set of parameter values to produce a set of theoretical predictions (see Figure 3B).”

2.30) 122] Do you mean that you “fit a sinusoidal contact rate to the inferred monthly transmission rate and obtain estimates of amplitude and phase”?

This has been deleted as part of the updated results.

Referee: 3

We thank the Referee for their useful comments.

We want to point out a major change upfront based on comments from yourself and Referee 2: we have changed the inference model to be a sinusoidal transmission function as opposed to a six-point spline. We believe this not only simplifies the description of the fitting process, but it indeed strengthens the results. We also want to point out that we have changed the y-axis of Figure 4 to be “percent synchronized” instead of “percent out of phase” (terms explained below and, in the MS).

This manuscript presents data and modeling analyses showing a surprising diversity of measles dynamics for different regions within the city of London, during a decade in the prevaccination period. Despite patterns that appear strongly annual at the aggregate city-wide level, the authors show that dynamics within each of nine sub-regions divide between annual, biennial, and a few that have less distinct period. This empirical finding is an interesting insight into city-scale measles dynamics, which as the authors note have been a canonical success story of simple mass-action models, which assume random mixing and average over any internal

heterogeneities. This is inherently interesting and has potential to advance our understanding of basic principles of disease dynamics. The authors analyze the data using a series of modeling and statistical approaches. Their major findings center on the role of seasonality in transmission in driving the observed differences in periodicity among sub-regions, and on the influence of coupling between sub-regions on synchrony/asynchrony. They conclude that they can explain the diversity of dynamics on theoretical grounds, and they draw parallels to other recent studies of measles or theoretical disease dynamics.

I am joining this review process after a previous round of review and resubmission was completed. I read and considered the ms in its entirety, and formed my opinion, then read the response to reviewers to see if it shed any light on my outstanding questions and concerns. Below I will summarize these, making minimal reference to the previous reviews and responses.

On the whole I was very intrigued by the dataset, and agreed with the broad outline of the authors' approach to analyzing it. The conclusions seemed plausible, at a qualitative level, and the framing in the introduction and discussion was excellent. However there were numerous issues in the methods and results that raised concerns for me, including many issues of clarity/communication that I was surprised to see in a resubmitted manuscript. As a result I could not quite get on board with the repeated claims of strong agreement between theory and data. I summarize my concerns below.

We thank the referee for joining the process and their suggestions. We hope we can provide satisfactory explanations.

3.1) A basic description of the data source was missing, and the brief mentions throughout the ms were slightly confusing. The paper is initially framed in terms of "borough mortality records" (line 66) but it isn't explained how these relate to measles cases. Is measles listed as the cause of death? Is this unambiguous, always? Line 86 mentions "measles notifications", which sounds like a different data source from borough mortality records. The caption to Figure 1A says "monthly measles mortality reports", while 1B says just "Mortality". I am guessing all of these refer to the same data source, but am uncertain, and a bit unclear what that data source actually is.

We have clarified in Line 66 that these refer to borough-specific measles mortality records. We have changed notifications to be "mortality records". We have further clarified Figure 1, and note that we are only analyzing mortality records in which measles is listed as the cause of death from the Registrar General's reports.

3.2) The primary result of the modeling analysis is the conclusion that local variation in seasonality drives the variation in periodicity among regions. The chief finding from the data analysis is shown in Fig 3A, which superimposes the inferred seasonal transmission curves for all regions. Then these findings are related to the theoretical predictions in the bifurcation diagram in Fig 3B, which shows how different amplitude and phase parameters gave rise to different periodicity in a deterministic simulation model. The broad contours of this result are interesting, but I was left with lots of questions. The most important of these was how the

seasonality models used in the two analyses relate to each other. Fig 3A uses a spline-based model, which leads to the (curious) double-peaked functions shown in the figure. But the analysis in Fig 3B uses a sinusoidal curve with period 1 year. How well can this function capture the empirical patterns from the data? The text (lines 121-123) acknowledges this was an approximate fit, but the fit is never shown, and no uncertainties are given for the fitted amplitude and phase parameters. Particularly for the 2-year group (shown in blue), where the inferred seasonality looks almost like a sinusoid with a period of 0.5 year, I can't really imagine how a period-1 sinusoid could fit it. This undermined my faith in this core result of the ms.

To address this comment, we have reformulated the regional model formulation to use a sine transmission function. This makes the description of the fitting process more straightforward, but we also believe strengthens the results. A consequence of this is that the sine function eliminates any double peaks that were present in the previous set of analyses.

There were also more minor questions about this figure set, including:

- In Fig 3A, which region is in which periodicity group? Are any regions in the 2+ group? The black and grey colors are difficult to tell apart in the figure.

The periodicities are shown in Figure 1A for each region by color code. No regions are in the 2+ group. We have used a lighter grey here to help distinguish the 2+ group.

- why are there 13 months time-points for each series in Fig 3A? What is the meaning of the extra point after December?

We have clarified in the methods that we binned the mortality data to four-week intervals. Therefore, there are 13 approximate months, resulting in an extra point.

- Fig 3B shows ten separate points, representing the parameters of each region plus the aggregate city (which should be omitted, or shown with a different symbol). These are impossible to evaluate without seeing the inferred seasonality traces for each region. Indeed I had originally noted that Fig 3A should 'connect the dots' to show individual traces, but I gather from the response to reviewers that the authors deliberately removed these.

We have decided to leave the seasonality patterns shown as dots with the mean curve. However, now that seasonality is modeled as a sine function, the groupings as much closer as they are only based on two parameters. We hope this formulation is easier to interpret. The aggregate is now shown as a square in both A and B.

- It is never stated how exactly the periodicity is assessed for each time series (in the data or the deterministic simulations). From the power spectra shown in Figs S1 and S2, the period is sometimes ambiguous (e.g. outer South, outer West).

We have clarified that periodicity is calculated by taking the nearest-integer period corresponding to the maximum power via spectra analysis.

3.3) The second major result from the modeling analyses addressed the influence of coupling between regions. The outputs of this analysis were posed entirely in terms of whether the dynamics of two patches were in-phase or out-of-phase. I do not understand what this means in the context of the North-West pair. How can an annual signal (North) be out of phase with a biennial signal (West)? Similarly, on pp6-7 of the SI, there are strong statements about East-Central and East-Outer East all being strongly in-phase. What does this mean, since they're all annual? At the bottom line, I felt like I was unable to understand how the authors were talking about in-phase/out-of-phase throughout the manuscript. How were these terms defined, and how was the determination made for the quantitative analyses shown in Figs 4 and S6?

We have changed the wording here to be “synchronous” / “synchronized” where we clarify that synchrony in this work is defined as two regions with the same periodicity. We further clarify how the determination was made:

“For each simulation, we quantified the periodicity in each *in silico* region. If both were the same periodicity, we called them synchronized.”

Additionally, we have clarified how we determine periodicity, copied again below:

“We quantified the periodicity by taking the nearest-integer period corresponding to the maximum power via spectra analysis.”

3.4) The section on coupling rates made no mention of the importation rate, ι , estimated in the single-region analyses. There is a statement (lines 138-139) that local parameters were “inherited from the regional analysis” for the coupling analysis – did this include the importation rate? Also, the parameter estimates shown in the SI table show a large range of ι values estimates for the regions (from 4.71 to 80.43). How did these values relate to the pairwise coupling values estimated in Fig S7?

We have clarified that this includes the importation rate. We found no significant relationship between the importation and coupling rates.

3.5) Throughout the ms, I was struck by the many claims of ‘generally close fits’ and ‘strong visual fit’ and so on. This struck me as strong language which was not always borne out by the figures, e.g. Figs S3, S4 and S5 are not obviously “close fits”. Even Fig 2 is debatable, since the model does a good job with the big 1898 epidemic, the two-year cycle in the West region, and the general envelope of amplitudes, but otherwise there’s not much sign that the fitted model curves are correlated with the data. I noted the discussion of this point in the response document, and I’m sympathetic to the authors’ position. However I think their case would be stronger if they addressed the specific elements of the model fit that they believe support their case. I found the current phrasing (strong but vague claims of close fit) to distract and slightly undermine the claims that the theory explained the data.

In our revised version we have tempered the language, aimed to clarify what in particular we find compelling, and included more caveats. A couple of examples from the text follow:

In the results section:

“In particular, we note the fitted model was able to visually capture the periodicity, in particular the West region, and the general shape and size of epidemics across locations, in particular the larger 1898 outbreak. We further compared the predicted power spectra against the data (shown in the Supplement). While the majority of regions agreed well with the observed spectra, some such as the South and outer West departed from the observations, indicating that in those regions each stochastic simulation from the fitted model may not always be biennial.”

In figure 2 caption:

“The simulations, shown in light blue bands of 95% quantiles, generally visually match the observed data, shown in red, in terms of periodicity, outbreak size (particularly the 1898 outbreak), and general shape.”

In Figure S3 caption:

“In addition to visual fits, we also evaluated the fitted model on power spectrum. We find generally close agreement with the data for this metric when we examine the period corresponding to the peak of the maximum power. However, the fit in the South region differed from the data, producing a mean bias around the annual attractor, although many simulations were still biennial when examining the 95% quantile (shaded band). Other regional fits, such as the West resulted in the corrected periodicity, but a lower normalized power peak.”

Among other examples, we hope this revised language is more specific and reflective of the results.

3.6) There are numerous small technical errors throughout the manuscript.

- Line 118, allowing the phase parameters to take values from 0-2 doesn't correspond to a maximum 2-month lag. The expression is in units of radians, so it is $2/2 \cdot \pi$ years, or roughly 4 months.

Fixed as below:

“range: 0-2, corresponding to a maximum lag of approximately four months”

- Lines 132-134, $\sigma=0.01$ does not mean that the infected population in Y contributes 1% of infections in population X. This depends on the absolute values of I_Y and I_X . The sigma

gives a relative weighting to those contributions, which will scale with current prevalence on each patch.

We have clarified that this is at a given time as below:

“For example, a coupling rate of 0.01 between locations X and Y at given time is equivalent to assuming that the infected population in Y provides 1% of the infections in population X at this same time”

- Lines 143-144 and 215-216, $\sigma = 1$ does not correspond to a well-mixed system. This would be $\sigma=0.5$. $\sigma=1$ is a curious case where only I_Y contribute to infection on patch X. As a consequence, I think Fig 4CD should be restricted to the range $\sigma = 0$ to 0.5.

Fixed.

- Line 130, which N is used here? In the original Keeling & Rohani paper all patches had equal size, but here it matters.

We have updated the notation to be more precise as below:

“For each location, we modulated the force of infection to include a coupling parameter related to the number of infected in the other location ($\beta(I + \iota)/N$ to $\beta((I_X + \iota_x)(1 - \sigma) + I_Y \sigma)/N_x$ (for population X coupled with population Y, where ι_x refers to the importation rate into population X)).”

- As presented and defined in the SI, the modeling of the importation rate is wrong. This process, represented by ι , is defined as the ‘yearly infected importation rate’. This gives it units of cases over time. But in the model on page 1 of the SI, it must have units of cases as it is added to I (the number of cases) and multiplied by λ to contribute to the incidence of new infections.

Corrected to read: “ ι is the mean infectious importation rate”

- Even if this is treated as a minor semantic issue, it raises quantitative concerns since the table of parameter estimates on SI page 4 reports values up to $\iota = 80$ for individual regions. If this value is used in the equations, as written, it would be a major driver of the dynamics. If it is modified somehow, this needs to be explained. There is a vague statement about the stochastic version of the model, based on Euler transitions (a term I’m not familiar with. Is this equivalent to the Gillespie algorithm? If not, how is it different?). Regardless, this does not clarify how ι is treated since it has equivalent status to the state variable I.

We have clarified how ι is treated above. Since there have not been many analyses examining this era of historical data, it is possible that this high importation rate is a genuine feature of the population structure at this time. However, if we examine the import rate in the same manner as CBR, we find that phase and amplitude still explain

the majority of the variation in periodicity. Future work should examine temporal variation in import rates across eras.

We have added more explanation, and a reference, on the Euler-multinomial process.

- Line 103, what is a ‘dispersed normal distribution’? Reading the relevant part of the SI, should this be ‘discretized’?

We have clarified this as follows:

“we fit the stochastic SEIR model to each region using an observation process that allows for measurement error per (I3)”

- The description of the observation model (SI lines 28-39) is impossible to understand without reference to the cited source. There is an error in the first equation (final term lacks proper exponents) and the notation is sloppy (use of comma for semicolon, so I was initially confused about how the normal distribution could have 3 parameters). In lines 30-31, why is rho set equal to 1, and what is ‘measurement error’ given that under-reporting is included separately? Is it for false positives? In line 36, does ‘mortality’ refer to true deaths or observed deaths?

We have corrected the notation and added explanatory text as follows:

“To fit the model to the observed weekly mortality data, *deaths*, we modelled true deaths (*D*) as the sum of $CFR\gamma I$ in a monthly period, where CFR is the inferred case fatality rate. We then model observed deaths, (i.e. *deaths*) via an over-dispersed binomial model per (*I*) with measurement error (e.g. false positives) (ψ). Although under-reporting could be estimated in this framework, we choose to fix it at 100% here as we assume mortality is well-reported.

$$P[deaths | D] = \Phi(deaths + 0.5; D, \psi^2 D^2) - \Phi(deaths - 0.5; D, \psi^2 D^2)$$

”

- SI line 22, this is insufficient detail for me to understand how the cubic spline fitting was done.

As seasonality is now modelled as a sine function, there is no longer any cubic spline fitting performed.

Minor points

- Several symbols are used for multiple quantities in the ms. phi is used for both the case fatality rate (SI page 1, table on page 4) and the phase of the sinusoidal transmission function (SI page 8, Fig 3)? sigma is both the inverse of latent period and the coupling rate.

Apologies -- fixed.

- I found it strange that results for the aggregate city-wide data were mixed in with the results for the regions. In figs 3A and 3B, this would be easily remedied by using a different symbol for the aggregate data. At other points, e.g. on lines 116-117 where the mean transmission rate used for simulations seems to be an average of the aggregate parameter estimate averaged with the regional estimates, I would suggest it should be removed.

Fixed. The mean transmission rate is now just from the nine regions. Additionally, the aggregate is now shown in square points in A and B.

- Line 99 – relative to what? Why then is it appropriate to use the quantitative estimates to claim support for the theory (lines 176-179), without caveat?

We have clarified this is relative to imperfect reporting:

“To estimate the time-invariant case fatality rate (CFR) (27), we assumed perfect reporting of measles mortality, hence this is somewhat of a relative measure in the case of imperfect measurement.”

We have added the following caveat:

“For example, our estimated CFR (ranging from 0.9% to 1.7%), compared well with a meta-analysis by (27), in which the observed nationally-representative CFRs fell between 0.23%-1.51%, further confirming our ability to confront theory with data, albeit we assumed complete reporting accuracy in our model.”

- Very little information given about the iterated filtering. How was convergence assessed? How many runs were done?

We have added this information in the main text as:

“This process is repeated many times (here, we used 400 unique parameter sets) to acquire estimates of the likelihood for each parameter set, as well as fully explore the multi-dimensional parameter space for 60 iterations”

We have also added an example diagnostic showing convergence to the supplement showing convergence of log likelihood.

- Line 151 – it would be helpful to give all population sizes somewhere.

Population sizes have been added to the supplementary table with parameter values.

- Line 158 – what is the basis for this statement about susceptible population sizes? It seems to hinge on attack rates, ages of infection, etc, and I could not work all this out in my head.

We've deleted this statement as it was not clear and did not add any additional understanding.

- Line 175 – I couldn't understand what this ('indicating potential periodic variation' was trying to say).

We have clarified this to mean that the simulations for the fitted model may not always be biennial simulations as below:

“While the majority of regions agreed well with the observed spectra, some such as the South and outer West departed from the observations, indicating that in those regions each stochastic simulation from the fitted model may not always be biennial.”

- Line 210 – I was puzzled by the statement about differences in phase modulating the effective R_0 at the start of the year. I have since read the authors' explanation in their response to reviewers, which is more information but still not very convincing to me. It seems to me that this phase effect differs importantly from the effect of population susceptibility, because it is transient and will average out over the course of the season, whereas the effect of susceptibility applies to the whole season.

We have reworked this sentence to indicate merely that there is evidence that subtle changes in seasonal contact can produce variable periodicities as below:

“Although the impact of seasonal phase on periodicity may seem counterintuitive, variation in this parameter, subtle differences in seasonal forcing have previously been shown to modulate periodicity (Dalziel et al 2016).”

- Line 260 – in what way are the findings consistent with Keeling & Rohani's theory, which predicts that synchrony rises with coupling up to a critical value, then declines beyond that. There is perhaps a hint of this in Fig 4C?

We have removed this sentence in light of the updated analysis.

- Line 293 – is this statement consistent with the earlier discussion about the decline of synchrony beyond a critical coupling level?

We have removed the prior statement in light of the updated analysis.

Appendix C

Response to Referees for Manuscript RSPB-2019-1510.R2 entitled "Coexisting attractors in the context of cross-scale population dynamics: measles in London as a case study"

We greatly appreciate the opportunity to respond to this additional round of reviews for our manuscript, "Coexisting attractors in the context of cross-scale population dynamics: measles in London as a case study".

The manuscript has now had multiple iterations, and we have substantially incorporated this feedback in both our analyses and framing of the results. In particular, we have removed references to "synchrony" throughout the manuscript while opting for "same periodicity" when appropriate. We have also changed our references to "bifurcation diagram" to "dynamic grid". Additional points of clarity have been incorporated into the main text, in particular the discussion section.

We would like to thank you, the Associate Editor, and the reviewers for their careful, considered, and very helpful comments. We are additionally thankful to have a chance to resubmit our manuscript. In the letter that follows, we will respond to comments and outline our changes.

We appreciate your consideration of our work for publication and the chance to resubmit our manuscript. Please let us know if you have any additional questions or concerns and we look forward to hearing from you.

With best regards,
Alex Becker, Susan Zhou, Amy Wesolowski, Bryan Grenfell

On the plus-side, the new version shows significant convergence; on the minus-side, doubts remain and new problems have been introduced in your revision. The latter concerns, for example, the use of 'synchrony'. I appreciate that you have made this change in response to earlier comments, but it is an unfortunate change. I feel you cannot adapt a word that has a clear interpretation in the mind of the reader and redefine it for your use. Clearly, synchrony is more than having the same periodicity. Using that word is therefore confusing and suggestive of more than can be justified by the results. Several other issues need to be mentioned clearly in a discussion, for example, remark 2 by the Associate Editor and the remark on the CFR-estimates by the reviewer. Although you should discuss remark 2 on the interpretation of Figure 3B, I also agree with the Associate Editor that it seems strange to call 3B a bifurcation diagram. There is no system state of which the stability-regime is assessed as a function of a changing system ingredient. Please clarify or change the way you refer to the plot. All other remarks should be addressed, but mainly asks for some additional clarification and some minor rewriting.

As you have realized by now, we have made an exception in your case to allow another round of revision (reason: we like the work, but we have not yet reached the level where everyone is unconditionally convinced of its findings). However, if profound doubts remain this could still lead to rejection. So I urge you to make every effort to fully address all of the comments at this stage. If deemed necessary by the Associate Editor, your manuscript will be sent back to one or more of the original reviewers for assessment. If the original reviewers are not available we may invite new reviewers. Please note that we cannot guarantee eventual acceptance of your

manuscript at this stage.

Associate Editor

Board Member: 1

Comments to Author:

Two key points from me:

1) The referee raises the use of the word "synchrony", and I strongly agree: to say two periodic things were in synchrony must surely mean both the same period and the same phase, i.e. they happen at the same time. When it's only known to be the same periodicity, it does not follow that they are synchronous. This should be resolved in the wording throughout (and not by redefining the word "synchrony" see some instances below).

We have changed references to synchrony to be “same periodicity” there are some parts where this may result in clunky language, but given the prior changes, this may be unavoidable. There is the option of “coperiodic” as well which may be slightly better, however given that it does appear to have a specific mathematical meaning and is not a commonly used phrase, we are reluctant to include it in this revision.

2) The result that the dynamic regime depends on phase is still puzzling (figure 3B - which I don't think really is a bifurcation diagram). If it's just a seasonally forced SEIR model then the system should be invariant to phase - the whole thing should shift so if we're measuring rounded integer year of period, surely it should be invariant (fig 3b should be vertically constant). Is there something else hidden which is periodic (can't tell from supplement - are births a function of time), or could this just be an artefact of initial conditions or how periodicity is measured?

We have changed mentions of “bifurcation grid” to be (primarily) “dynamic grid” throughout the manuscript. We clarify here that while the long-term dynamics should be invariant to phase, the short-term dynamics are not. As phase was identified as a key driver in the PCA analysis, we have chosen to model this situation.

We have added this to the Discussion (Line 354) as below:

“The close visual fits to the data shown in Figure 2 inspire confidence in the inferred model’s ability to accurately describe the underlying dynamics, thus allowing us to perform detailed comparison with theory. Similar to (9), we find that subtle differences in seasonal transmission structure can lead to different periodicities. Using a dynamic grid based on this variation in seasonal amplitude and phase, we provide a plausible explanation for the array of observed data based on measurable differences in seasonality. While long-term deterministic dynamics become invariant to seasonal phase, shorter-term, non-equilibrium, dynamics display rich heterogeneity as exhibited in Figure 3B. As seasonal phase was identified as a key parameter in a PCA analysis, we chose to model this likely transient-driven dynamic. Future work aims to quantify and better understand these ecological transients sensu (32).”

Some smaller points from me also, working in order through the current manuscript, mostly

small typos or unclear wording resulting from previous edits:

line 75: "role of demographic on" -> demography surely

Corrected.

90: "based their" missing an "on"

Corrected.

116: "forwarded simulated" surely "forward simulated"

Corrected.

124: "regions locales", one or the other I guess

Corrected.

135: see above, this doesn't make sense as a definition of synchrony consistent with others' usage.

Changed to "To investigate how variation in regional periodicity"

144-146: some awkward wording (edits not helped): "at given time" so coupling value is a function of time? "provides 1% of the infections in population X" isn't quite parsing right. Looking at the model, it's β of the infecteds act on the force of infection at the other place, so it's not 1% of infections, but maybe 1% of the new infections or infection rate.

Apologies – we meant that at a given time β is contributing that percent of the new infections, so β is constant but the contribution of $\beta * I$ is not. We have rephrased this to be hopefully clearer:

"For example, a coupling rate of 0.01 between locations X and Y is equivalent to assuming that the infected population in Y contributes 1% of the new infections, albeit scaled by β/N , in population X"

154: "synchronous (i.e. same periodicity)"

Changed to be "the variable periodicity North – West pair and the both biennial West – outer West pair."

191-194: not clear: ".. in those regions each stochastic simulation... may not always biennial..".
? Maybe "not all stochastic realisations were biennial"?

Thank you, we have changed this to be "indicating that in those regions not all stochastic realizations of the fitted model were biennial."

228: *another i.e. redefinition*

Changed to be “Thus far we’ve analyzed local dynamics to gain insight into the drivers of region-specific measles periodicity; we now turn to the potential role of population coupling on producing metapopulation dynamics of the same periodicity between two regions.”

241: *another redefinition of synchronised.*

Changed to be consistent with above.

247: *synchrony through this paragraph*

Changed to use “same periodicity”.

Reviewer(s)' Comments to Author:

Referee: 2

Comments to the Author(s)

The authors document measles time series of diverse dominant periodicity for London regions, estimate important model parameters from the data with the aid of demographic information, and reproduce to some extent the observed patterns by incorporating specific coupling rates and sinusoidal phase and amplitude transmission estimates into the model. The data set analysed is rather unique and their findings interesting to the field, but I strongly discourage the use of the term “synchrony” to refer to a coincidence in interepidemic period, simply because they are not interchangeable: biennial time series, for example, may present outbreaks in alternate years and thus be in opposite phases, a situation we would like to distinguish from biennial epidemics that peak simultaneously. Moreover, the authors later make reference to previous work that examined changes in synchrony among boroughs and cities as measured by the correlation between time series, further demonstrating that calling different notions the same leads to confusion.

Please see above response to comment 1 by the Associate Editor.

Another aspect that could be discussed further is why is the importation rate assumed to be constant, when a fraction of the number of infectious individuals in the other patch at each time step could be used instead.

We agree with this point. We have added a sentence to the discussion on Line 311 which reads:

“Additionally, although we assumed a constant importation rate in our local analysis, a fraction of infected individuals in other populations (i.e., a full metapopulation model) could provide an alternative approach.”

Related to that, the text in [146] states “we used the seasonal transmission values” which is not the number of infectious individuals in the other population and I therefore do not find justified.

We have clarified we use the local parameters:

“To parameterize the two-region model, we used the parameters inferred from the region-specific monthly data (e.g., seasonal transmission values) with the goal of estimating pairwise regional coupling.”

We listed beta initially as we indicate we go from $(\beta(I + I_x)/N$ to $\beta((I_X + I_x)(1 - \sigma) + I_Y\sigma)/N_x$ earlier in the paragraph so we wanted to keep the notation / framing consistent, but we agree that it was confusing as written.

It does not make much sense to me that forcing phase plays the same role as initial conditions, could the authors provide more detail to support this argument?

We have added an explanation to the Discussion that these dynamics are likely transient dynamics, however as phase was found to be important in a PCA analysis, we have chosen to model these dynamics. Future work aims to understand them further.

“The close visual fits to the data shown in Figure 2 inspire confidence in the inferred model’s ability to accurately describe the underlying dynamics, thus allowing us to perform detailed comparison with theory. Similar to (9), we find that subtle differences in seasonal transmission structure can lead to different periodicities. Using a dynamic grid based on this variation in seasonal amplitude and phase, we provide a plausible explanation for the array of observed data based on measurable differences in seasonality. While long-term deterministic dynamics become invariant to seasonal phase, shorter-term, non-equilibrium, dynamics display rich heterogeneity as exhibited in Figure 3B. As seasonal phase was identified as a key parameter in a PCA analysis, we chose to model this likely transient-driven dynamic. Future work aims to quantify and better understand these ecological transients sensu (32).”

Minor points:

[28] What is meant by “phase coupling”?

We have changed this to be “epidemic coupling”.

[89] The statement “the outer ring of London was not formally divided into regions” makes me wonder whether the mortality records used in this study were reported in the groupings used in this paper, or whether they were grouped in these regions by the authors. Which was the case?

We grouped these regions based on their present-day location and the nearest inner region.

[95] What resolution in time is assumed for the interpolated data? How are the interpolated data employed?

We have clarified that these data were interpolated to the monthly scale. These data are then fed into the model as known covariates (e.g., births).

[114] The authors do clarify in their response to referees that they performed 60 simulations for each of the 400 sets of parameters used, but this not as clearly stated in the paper.

We have clarified this to read: “This process is repeated many times (here, we used 400 unique parameter sets) to acquire estimates of the likelihood for each parameter set, as well as fully explore the multi-dimensional parameter space for 60 iterations for each parameter set”

[123] Is the transmission rate “based on” or equal to the mean of their estimates?

We have clarified that it is equal to the mean.

[138] Could you provide more details as to how was regional isolation “relative” in the previous model? Were modelled populations not completely independent?

We have removed the word “relative” here as regions were modeled completely independently.

[182] How is population size related to interepidemic period (other than by defining whether the population is above the CCS)?

Changed to read “Surprisingly, the observed variation in CBR did not correlate with the observed periodic signal.”

[196] I am unconvinced of the source used for CFR estimates. The cited paper (27) uses data from 1980 onward, while the paper under review analyses a data set from 1897-1906. Why would the CFR be similar in such different eras? It has definitely gone down since the turn of the 20th century, and in the 1980s plenty of treatments and medical knowledge were available that were not over eighty years earlier.

We have updated the text to read:

“For example, our estimated CFR (ranging from 0.9% to 1.7%), compared well with a meta-analysis by (27), in which the observed nationally-representative CFRs fell between 0.23%-1.51%. Early 1900’s estimates of CFR for London are challenging to assess as measles cases were not yet notifiable and therefore the incidence reporting rate is unknown for this era, however, the reported CFR for Paddington from 1904-1906 was 2.7%-4.3% (31). Crudely applying the incidence reporting rate of approximately 50% in the later era (13) yields an approximate CFR of 1.35%-2.15%, in line with our estimates here. However, we assumed complete reporting accuracy in both our model and historical rate correction. In the case of imperfect reporting, the estimated CFR would likely increase.”

[198] Please discuss how incomplete reporting would modify these estimates.

We have clarified that “In the case of imperfect reporting, the estimated CFR would likely increase.”

[234] Please include references to the birth rates of each region in your discussion.

We have added a sentence to the Discussion (Line 342) which reads:

“Likewise, differences in socioeconomic conditions may have manifested themselves in the substantial range of crude birth rates observed across the nine regions spanning ten years (22 to 52), however this variation, while still important to the overall dynamics, was not identified as a primary driver of periodicity.”

[277] What is meant by a “bifurcation grid”?

We have changed this to be “dynamic grid”.

[304] I would expect lower school attendance to lower forcing amplitude, rather than modifying its phase.

We’ve removed this sentence and condensed the references to (9) in this paragraph.

Appendix D

Response to Referees for Manuscript RSPB-2019-1510.R2 entitled "Coexisting attractors in the context of cross-scale population dynamics: measles in London as a case study"

We would like to thank you, the Associate Editor, and the reviewers for their careful, considered, and very helpful comments. We are additionally thankful to have a chance to resubmit our manuscript. In the letter that follows, we will respond to comments and outline our changes.

We appreciate your consideration of our work for publication. Please let us know if you have any additional questions or concerns and we look forward to hearing from you.

With best regards,
Alex Becker, Susan Zhou, Amy Wesolowski, Bryan Grenfell

Referee: 4

This manuscript is quite interesting in terms of both epidemiology and complex system. Previous studies have been pointed out that the possibility of coexistence of multiple attractors in the system which the force of infection is seasonally varied, this paper suggests example observed in real world. I really enjoyed reading except one question: the authors showed legion specific periodicity because of the estimate of coupling rate is small. I am a bit surprised that biannual oscillation was observed for a long time even though immigration events (equivalent with coupling) should have happened. If considerable explanation can be existed, please discuss.

We thank the Reviewer for their positive evaluation and feedback.

A likely explanation is that although importations did occur, the attractor was predominantly dictated by seasonal forcing in tandem with low coupling values. From London-specific data in 1944-1966, we do know that the biennial attractor faded away and all of London was annual, at least until the Baby Boom in 1950, where each region of London rapidly switched to a biennial cycle. However, as London-specific data unfortunately do not exist for the entire continuous period of 1897-1944 (as far as we are aware) it is challenging to assess precisely how the coexisting attractors faded from London.

We have included a comment on this in Line 280 of the discussion, as follows:

“Additionally, although we assumed a constant importation rate in our local analysis, including a fraction of infected individuals in other populations (i.e., a full metapopulation model) could provide an alternative approach that would allow for the disentanglement of internal coupling and external importations”